# Coordinated representational reinstatement in the human hippocampus and lateral temporal cortex during episodic memory retrieval

D. Pacheco Estefan[1,2], M. Sánchez-Fibla [2], A. Duff[1], A. Principe[3,4], R. Rocamora[3,4,5], H. Zhang[6], N. Axmacher[6] & P.F.M.J. Verschure[1,7,8]

Theoretical models of episodic memory have proposed that retrieval depends on interactions between the hippocampus and neocortex, where hippocampal reinstatement of item-context associations drives neocortical reinstatement of item information. Here, we simultaneously recorded intracranial EEG from hippocampus and lateral temporal cortex (LTC) of epilepsy patients who performed a virtual reality spatial navigation task. We extracted stimulus-specific representations of both item and item-context associations from the time-frequency patterns of activity in hippocampus and LTC. Our results revealed a double dissociation of representational reinstatement across time and space: an early reinstatement of item-context associations in hippocampus preceded a later reinstatement of item information in LTC. Importantly, reinstatement levels in hippocampus and LTC were correlated across trials, and the quality of LTC reinstatement was predicted by the magnitude of phase synchronization between hippocampus and LTC. These findings confirm that episodic memory retrieval in humans relies on coordinated representational interactions within a hippocampal-neocortical network.

[1] Laboratory of Synthetic Perceptive, Emotive and Cognitive Systems (SPECS), Institute for Bioengineering of Catalonia (IBEC), 08028 Barcelona, Spain. [2] Department of Information and Communications Technologies, Universitat Pompeu Fabra, 08018 Barcelona, Spain. [3] Epilepsy Monitoring Unit, Department of Neurology, Hospital del Mar, 08003 Barcelona, Spain. [4] Hospital del Mar Medical Research Institute, 08003 Barcelona, Spain. [5] Faculty of Health and Life Sciences, Universitat Pompeu Fabra, 08003 Barcelona, Spain. [6] Department of Neuropsychology, Institute of Cognitive Neuroscience, Faculty of Psychology, Ruhr University Bochum, 44801 Bochum, Germany. [7] The Barcelona Institute of Science and Technology (BIST), 08036 Barcelona, Spain. [8] ICREA, Institució Catalana de Recerca i Estudis Avançats, Passeig de Lluís Companys, 23, 08010 Barcelona, Spain. These authors contributed equally: N. Axmacher, P.F.M.J. Verschure. Correspondence and requests for materials should be addressed to P.F.M.J.V. (email: pverschure@ibecbarcelona.eu)

nfluential theories of human episodic memory have proposed that unique aspects of an experience are represented in cortical areas and bound together in the hippocampus[1–7]. A key prediction of these accounts is that during successful retrieval of associative memories a necessary interaction between the hippocampus and the neocortex must exist. Specifically, this concept of episodic memory proposes that detailed sensory representations of individual items rely on the neocortex whereas the hippocampus realizes an index to these cortical representations[8] through item-context associations[9,10]. In addition, these theories predict a distinct information flow from neocortex to hippocampus during initial memory formation and in the reverse direction during retrieval[11–13]. Here we set out to empirically assess these predictions by analyzing the electrophysiological activity from the temporal lobe of human epilepsy patients.

While there is now abundant evidence for the general involvement of neocortical areas and of the hippocampus during long-term memory encoding and retrieval[1–7], the neural mechanisms underlying the representation of the specific content integrated in episodic memory have only started to be addressed. At a cellular level, content-specific "engrams cells" were identified in the rodent hippocampus[14,15] and neocortex[16–17]; at the system level, multivariate analysis methods such as pattern classification[18] and representational similarity analysis[19] have been used to identify the representation of specific events (see ref. [12] for a review). Intracranial EEG (iEEG) recordings in epilepsy patients offer a unique opportunity to directly track the electrophysiological organization underlying content-specific representations and inter-regional information transfer at a fast time-scale[20,21]. Indeed, previous iEEG studies have shown that remembering an episode requires the reinstatement of a dynamical oscillatory state, the 'neural fingerprint' of a specific experience[22]. While some studies have reported the reinstatement of distributed oscillatory patterns across multiple brain regions[23–25], others have focused on local time/frequency patterns captured at specific sites[22,26,27]. This latter approach has been used to characterize the specific representational features of hippocampal and neocortical reinstatement, demonstrating the increased involvement of the former in the retrieval of contextual as opposed to item-specific memories[22]. A link between the quality of memory and reinstatement of item-specific information in the neocortex has also been established[23,25,28]. However, no previous study has simultaneously tracked hippocampal and neocortical reinstatement in humans, and therefore, whether these are dissociated in terms of representational features (i.e., contextual versus item-specific respectively, as previous research suggests), is currently unknown. In addition, differences related to the role of specific frequencies as well as the relative timings of reinstatement in the hippocampus and the neocortex have not been thoroughly investigated. Finally, while theories predict an interaction of the human hippocampus with neocortical areas during episodic memory retrieval, their coordinated or independent engagement in the reinstatement of episodic memories still remains to be established.

A critical mechanism that could underlie such coordination is the synchronization of oscillatory phases across brain regions[29]. Indeed, phase synchronization – in particular in the gamma range (30–100 Hz) – is generally thought to enable neural communication and information transfer in the brain[30,31]. However, a direct link between gamma phase synchronization and representational reinstatement during episodic memory retrieval has not been demonstrated before.

To address these issues, we analyzed the electrophysiological activity from the hippocampus (HC) and lateral temporal cortex (LTC) of human epilepsy patients ($N = 11$) that were laterally implanted with intracranial EEG electrodes (Supplementary

Table 1). We considered the LTC given its involvement in representing item-specific information during recognition memory[26,32,33] (Fig. 1a). In our experiment, participants performed a previously established Virtual Reality (VR) active navigation task involving a recognition memory test[34]. This paradigm compares memory for items either in their original spatial contexts (congruent condition: same room and wall position at encoding and retrieval) or in different contexts (incongruent condition: different room and wall position at encoding and retrieval), and thus provides a highly ecologically valid tool to probe episodic context-dependent memory relying on active exploration[34–36] (Methods; Fig. 1b, c). We first investigated the electrophysiological patterns supporting representational reinstatement of individual items and of item-context associations in HC and LTC separately, and then tested whether reinstatement was coordinated between brain regions and expressed in distinct phase synchronization. As in previous studies, representational patterns were constructed by concatenating epochs of time-frequency resolved power values (44 frequencies between 1 and 100 Hz) in windows of 500 milliseconds, overlapping by 400 ms[22,23] (Fig. 1d). Encoding-Retrieval Similarity (ERS) between feature vectors was calculated using Spearman's Rho (Methods). Our results show a dissociation of hippocampal and LTC reinstatement in terms of representational format, specific timings and contributing frequencies. Moreover, we demonstrate the coordination of hippocampal and neocortical reinstatement through gamma phase synchronization.

## Results

**Dissociation of representational reinstatement in HC and LTC.**
Our reinstatement analysis of the HC revealed a significantly greater ERS in congruent as compared to incongruent trials (ERS $_{same\ item,\ same\ context}$ > ERS $_{same\ item,\ different\ context}$; analysis was locked to the onset of item presentation; $p_{(corr)} = 0.026$; Fig. 2a, B). ERS occurred between ~1.6–3.1 s at encoding and ~0–0.5 s at retrieval, with a compression factor of around 3, consistent with previous reports of condensed time-warped reinstatement[28]. In contrast, we did not find any evidence for reinstatement of item-specific information in the hippocampus: ERS did not differ when the same item was encoded and retrieved (ERS$_{same\ item,\ all\ contexts}$) as compared to when one item was encoded and another item retrieved (ERS$_{different\ item,\ all\ contexts}$; all time windows or clusters, $p > 0.484$; Fig. 2c, left). The ERS item context congruency effect cannot be simply accounted for in terms of the differences between rooms (i.e., contexts): when we compared ERS involving the same or different rooms regardless of items (ERS$_{all\ items,\ same\ context}$ vs. ERS$_{all\ items,\ different\ context}$), we did not observe any significant difference (all clusters, $p > 0.077$; Fig. 2c, right). Hence, these results establish that the HC represents the binding of context and item information.

To elucidate whether differences in ERS were driven by increases in similarity during congruent trials or decreases during incongruent trials we analyzed ERS in the time window of interest for both congruent and incongruent conditions (Fig. 2d). We compared average ERS values in the identified congruent-incongruent cluster against chance, i.e., zero, separately for each condition. ERS values were significantly higher than zero in congruent trials ($t(7) = 8.772$, $p = 5.038e-05$) but did not differ from zero in incongruent trials ($t(7) = -0.784$, $p = 0.458$; note that this latter test is not circular because the cluster is defined by contrasting congruent vs. incongruent trials and not by comparing them individually to zero). We observed the same results when focusing only on correct trials (Supplementary Fig. 9), but not on incorrect trials (Supplementary Fig. 11A).

Representational reinstatement in the LTC showed a markedly different pattern (Fig. 3 and Supplementary Fig. 1). Contrasting

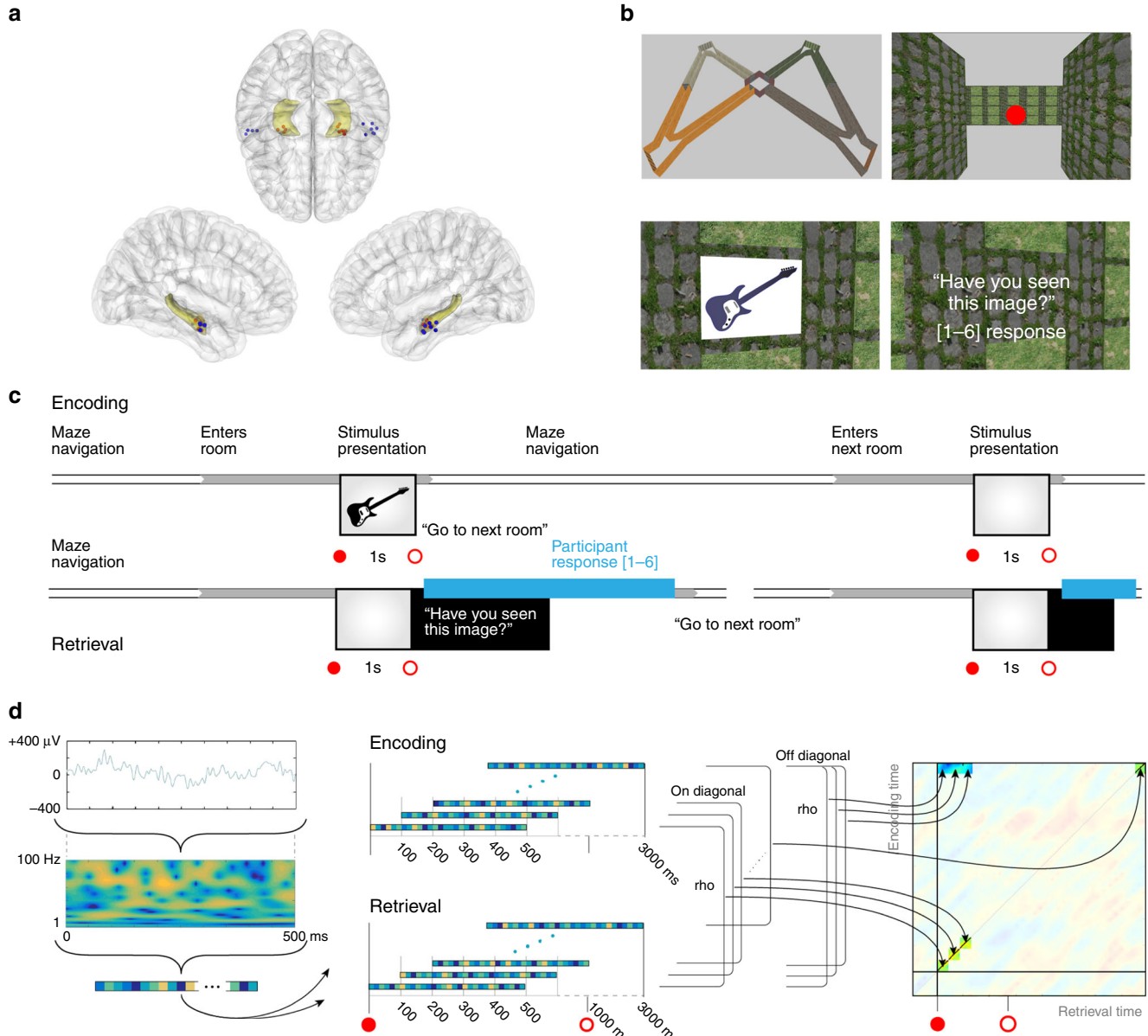

**Fig. 1** Task description and reinstatement analysis. **a** Electrode placement. One hippocampal and one lateral temporal cortex electrode from the same hemisphere was selected in each patient. Figure shows the selected contacts from all patients in MNI space. **b** Experimental task: overview. Participants navigated a virtual maze (top left) and visited several rooms (top right) where visual item were presented (bottom left). During retrieval, participants conducted a recognition memory task (bottom right). Items were either presented in the same room (congruent condition) or in a different room (incongruent condition). **c** Experimental task: individual trials. Timeline of one example trial during the encoding and retrieval blocks. **d** Representational similarity analysis. Time-frequency patterns were extracted from individual electrodes (44 frequencies between 1–100 Hz), in time windows of 500 ms, overlapping by 400 ms. Reinstatement corresponded to Spearman's correlations for all encoding-retrieval time pairs

the ERS values of the same item (regardless of context) between encoding and retrieval with ERS values of different items at encoding and retrieval yielded two significant clusters of context-independent item reinstatement ($ERS_{same\ item,\ all\ contexts}$ > $ERS_{different\ item,\ all\ contexts}$; cluster i: $p_{(corr)} = 0.013$, cluster ii: $p_{(corr)} = 0.018$; Fig. 3a). Specifically, we can distinguish two time windows during encoding (~0–1.1 s and ~2–2.6 s) that display distinct power frequency patterns that are reinstated during a retrieval time window between ~1–3 s. Average ERS in the item-specific or "same item" condition was significantly higher than zero in both clusters (cluster i: $t(10) = 6.326$, $p = 8.6135e-05$; cluster ii: $t(10) = 3.775$, $p = 0.004$), while it did not differ from zero in the "different item" condition (cluster i: $t(10) = 1.81$, $p =$

0.100; cluster ii: $t(10) = 1.071$, $p = 0.309$ Fig. 3b). Note that this same pattern of results was obtained when we restricted our sample of participants to those without ipsilateral hippocampal epilepsy ($n = 8$; see Supplementary Note 8 and Supplementary Fig. 19). In stark contrast to our results in the hippocampus, we did not observe any reinstatement of item-context associations (all clusters, $p > 0.48$; Fig. 3c). Also in this case, none of the clusters survived in the room-specific contrast (all clusters, $p > 0.109$; Fig. 3d). As in the hippocampus, we could replicate the main results when only including correct trials (Supplementary Fig. 10), but not when only including incorrect trials (Supplementary Fig. 11B). In additional control analyses, we investigated representational reinstatement at increased temporal resolution

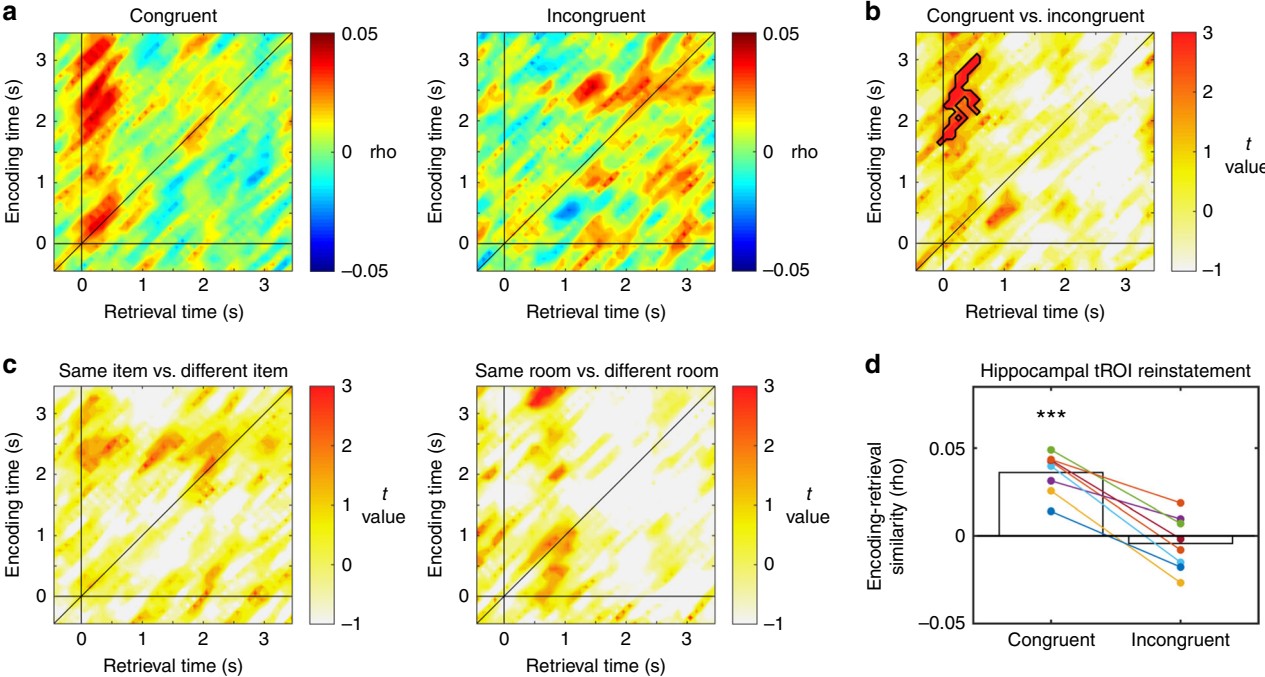

**Fig. 2** Reinstatement of item-context associations in the hippocampus. **a** Grand average reinstatement map for congruent (left) and incongruent (right) trials in the hippocampus, where the same item is viewed in either the same or a different context (room) during retrieval. **b** T-map of the congruent vs. incongruent contrast reveals a significant cluster at $p_{(corr)} < 0.05$ (outlined in black), indicating that HC reinstatement represents the binding of item-context information. **c** Reinstatement of same vs. different items regardless of context (left) and same vs. different contexts regardless of item (right): no significant clusters. **d** Reinstatement in the congruent and the incongruent conditions in the significant cluster in **b** tested against zero (one-sample t-test); colored dots indicate individual subjects. *** indicates $p < 0.001$

(Supplementary Note 4 and Supplementary Fig. 17), in several control areas (Supplementary Note 5 and Supplementary Figs. 4–6), separately in the different hemispheres (see Supplementary Note 6 and Supplementary Figs. 7 and 8) and locked to participant's responses during retrieval (Supplementary Note 9 and Supplementary Fig. 18).

To further corroborate our finding of item-specific reinstatement, we compared ERS of correctly remembered old items (hits) with ERS of correctly identified novel items (correct rejections; see Methods and Supplementary Note 1). We also compared high confidence hits with high confidence correct rejections. In the LTC, results revealed significant increases in ERS at very similar positions in time as those observed in our main analyses (Supplementary Fig. 13). In the hippocampus, no item-specific activity was observed (Supplementary Fig. 14), "but reinstatement" of hits versus correct rejections differed significantly between congruent and incongruent items (Supplementary Note 2 and Supplementary Fig. 15). Together, these results confirm the dissociation of hippocampal and LTC reinstatement in terms of representational formats.

**Functional relevance of LTC reinstatement**. We next assessed the behavioral relevance of reinstatement in HC and LTC. As expected, all subjects performed above chance in the recognition memory test (Area Under the Curve, AUC = 0.885 ± 0.14, $p = 1.988e-06$, paired t-test against 0.5). Previous literature has shown that contextual effects on recognition memory are small due to the salience of item information[34,37]. Given the highly accurate performance of 3 subjects in our test (with AUC values of 1, 0.985 and 0.977), and a generally higher variability in patients' performance as compared to healthy controls, we did not observe an effect of context in our group of patients: congruent and incongruent trials did not differ with regard to recognition accuracy ($t(10) = -0.171$,

$p = 0.863$) nor concerning subjective confidence ($t(10) = 1.579$, $p = 0.145$, Supplementary Fig. 2).

Since participants remembered most items, we focused first on the comparison between high and low confidence trials[38,39]. This contrast revealed higher levels of reinstatement for high as compared to low confidence trials in the LTC (ERS_high confidence, all items > ERS_low confidence, all items; $p_{(corr)} = 0.0104$), while no differential reinstatement in the hippocampus was observed (all $p > 0.260$, Fig. 3g). Please note that since reported confidence did not differ between congruent and incongruent conditions, these confidence ratings were not driven by the retrieval of contextual information in HC. Significant differences in ERS appeared in an encoding time window between 1.7 and 3 s and a retrieval time window from 0.5 to 2.5 s, i.e., before participants provided their responses (mean response time = 1.69 s, S.E.M. = 0.53 s, Fig. 3e). This effect substantially overlapped with the item-specific ERS cluster ii identified earlier (Fig. 3a). In this cluster, reinstatement of high-confidence trials was significantly larger than zero ($t(7) = 3.565$, $p = 0.009$), while it did not differ from zero for low-confidence trials ($t(7) = -1.847$, $p = 0.107$; Fig. 3f; note that this analysis is not circular because the cluster was defined by comparing high- vs. low-confidence trials rather than by comparing each of them individually to zero). The temporal overlap of confidence and item effects in the LTC suggests that the quality of item-specific reinstatement defines retrieval confidence. To corroborate this idea, we quantified mean ERS for high and low confidence trials in the clusters where we observed significant item-specific reinstatement (i.e., clusters i and ii in Fig. 3a). A direct comparison between the two types of trials revealed a significant effect in cluster ii ($t(7) = 3.619$, $p = 0.0085$, paired t-test), but not in cluster i ($t(7) = -1.64$, $p = 0.873$, paired t-test; Supplementary Fig. 3). However, comparing mean ERS values against zero revealed the same pattern of results in

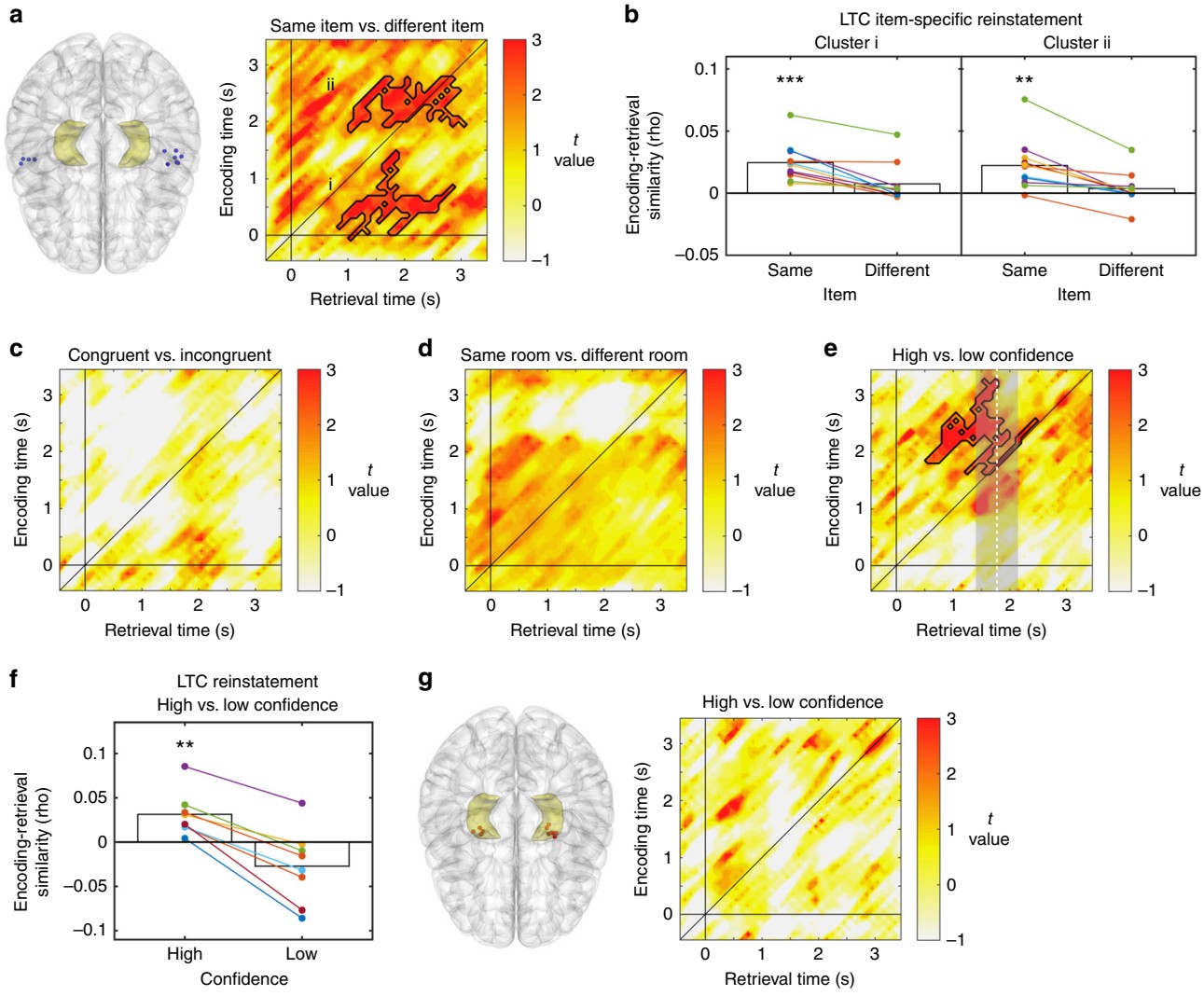

**Fig. 3** Reinstatement of item-specific information in the lateral temporal cortex. **a** *T*-map contrasting encoding-retrieval similarity for same items versus different items. Areas of significant correlations are outlined in black. **b** Mean reinstatement in these clusters for "same item" and "different item" trials. **c** and **d** *T*-maps for congruent vs. incongruent and same vs. different contexts (rooms) contrasts. **e** Encoding-retrieval similarity in the "same item" condition split into high and low confidence trials; *T*-map revealed a significant cluster. Dashed line and shaded grey area shows mean response time ± S.E. M across the group of patients. **f** Reinstatement for high and low confidence trials in the cluster observed in **e**, tested against zero. **g** Absence of significant confidence effects in the hippocampus. *** and ** in panels **b** and **f** indicate rho-values that are across the group significantly different from 0 at *p* < 0.01 and *p* < 0.001 respectively

both clusters, i.e., ERS values were significantly higher than zero in high confidence trials but not significantly different from zero in low confidence ones (cluster ii: high confidence, $t(7) = 4.504$, $p = 0.0027$; low confidence, $t(7) = -0.893$, $p = 0.401$; cluster i: high confidence, $t(7) = 3.07$, $p = 0.017$; low confidence, $t(7) = 1.12$, $p = 0.296$; Supplementary Fig. 3). We also directly compared LTC reinstatement in correct versus incorrect trials and observed numerically increased reinstatement for correct trials at the time where a confidence effect was observed (Fig. 3e). However, these differences did not survive correction for multiple comparisons (all $p > 0.525$; Supplementary Note 3, Supplementary Fig. 12). Taken together, these results support the notion that the reliability of stimulus-specific representations modulates retrieval confidence.

**Low-frequency dependence of hippocampal reinstatement.** To evaluate the specific contribution of the low-frequency bands to reinstatement in the HC and the LTC, we performed a jackknife

procedure – i.e., we recalculated reinstatement of item/context associations in HC and of item and confidence information in LTC after removing activity from Delta (1–3 Hz), Theta (4–8 Hz), and Delta-Theta (1–8 Hz; Methods). Note that this analysis is sensitive to the particular representational forms of each region, i.e., it assesses the reduction of context reinstatement in the hippocampus and of item and confidence effects in the LTC in their corresponding temporal regions of interest. We found that removing 1–8 Hz activity significantly reduced HC reinstatement in the congruent/incongruent cluster ($t(6) = 2.45$, $p = 0.049$, $t$-test against zero) but not in any of the LTC item-specific clusters or the confidence-related cluster (all $t < 1.42$; all $p > 0.202$, $t$-test against zero; Fig. 4a). This effect was not observed for Delta ($t(6) = 1.759$, $p = 0.129$, $t$-test against zero; Fig. 4b) or Theta ($t(6) = 1.3021$, $p = 0.24$, $t$-test against zero; Fig. 4c), suggesting a more relevant contribution of combined Delta-Theta activity for hippocampal than for LTC reinstatement.

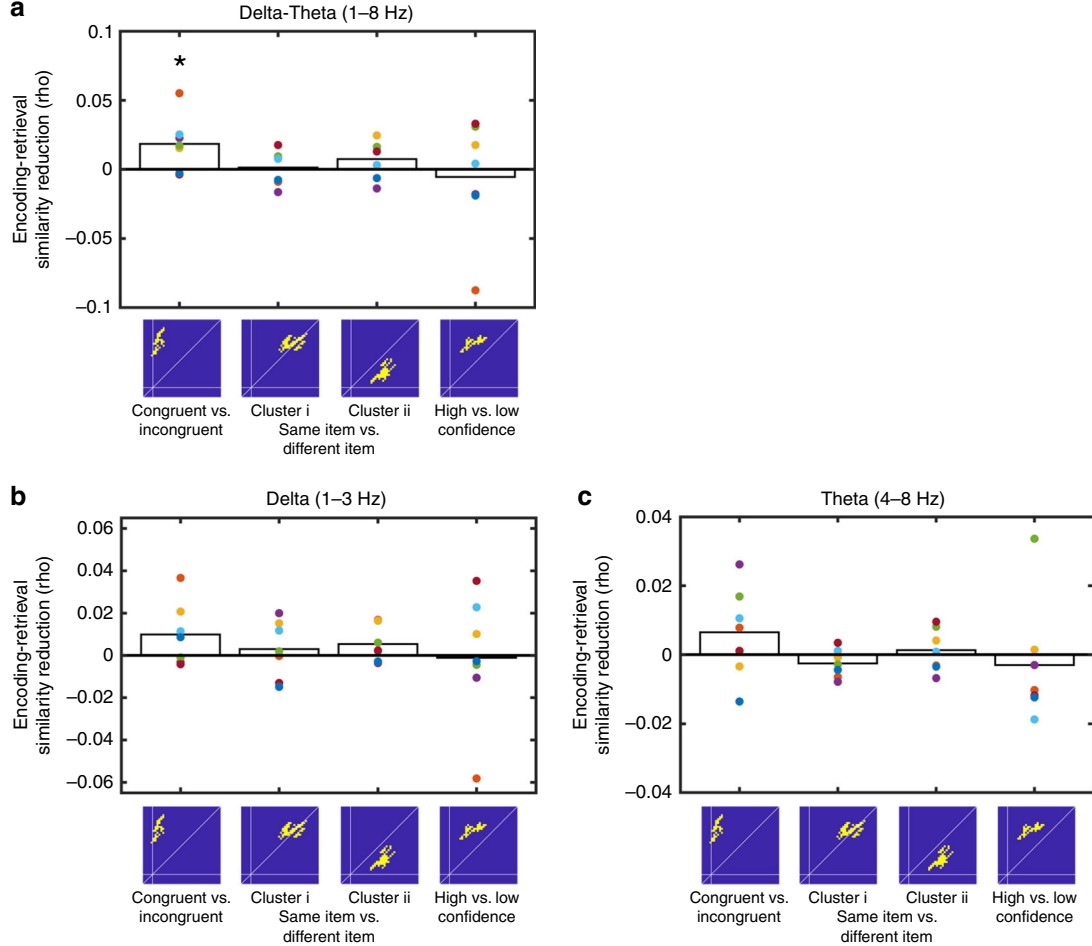

**Fig. 4** Frequency-specific reinstatement in HC and LTC. **a** Results from jackknife procedure showing significant reduction of hippocampal context reinstatement after excluding 1–8 Hz activity, while item-specific reinstatement and confidence effects in lateral temporal cortex were not affected. Bar plots indicate changes in encoding-retrieval similarity in the main contrasts and clusters after excluding 1–8 Hz activity. * in panel **a** indicates reductions in ERS values that are significantly different from zero across the group at $p < 0.05$. **b, c** Reduction of encoding-retrieval similarity in relevant temporal regions of interest after the removal of delta (1–3 Hz; panel **b**) and theta (4–8 Hz; panel **c**) activity. While reductions were numerically higher in the hippocampal congruent-incongruent cluster, no significant differences were observed

**Coordination of hippocampal and LTC representational reinstatement.** Our results thus far indicate that HC reinstatement of item/context association occurs during early retrieval time windows, < ~0.6 s, and that it is strongly dependent on low-frequency 1–8 Hz oscillations. HC reinstatement is followed by behaviorally relevant LTC reinstatement of item information in later retrieval time windows, > ~1 s, that relies on broadly distributed frequencies. We tested for a possible coordination of reinstatement between LTC and HC, by correlating the magnitude of LTC and HC reinstatement across trials. We focused on the encoding/retrieval time clusters that showed item-context reinstatement in HC (Fig. 2) and behaviorally relevant item reinstatement in LTC (Fig. 3), respectively. Because coordinated reinstatement is more likely to occur when it reflects activity from the same encoding time, we defined temporal regions of interest (tROIs) around the HC and LTC clusters aligned by their corresponding encoding times (Fig. 5a, Methods). As expected, the resulting temporal regions of interest showed the same effects as the original clusters: item-context reinstatement in the HC tROI, $t(7) = 8.974$, $p = 4.3449e{-}05$; confidence reinstatement in the LTC tROI: $t(7) = 3.973$, $p = 0.005$).

We calculated the correlation of reinstatement values across trials. Starting with the most global analysis, we first included all trials in the ERS$_{\text{same item, all contexts}}$ condition. We observed that correlations were significantly higher than zero ($t(6) = 3.353$, $p = 0.0153$) at the group level, indicating that trials in which hippocampal reinstatement was high also showed elevated LTC reinstatement (Fig. 5a, left). This effect was not observed when restricting analysis to the subset of trials in which hippocampal reinstatement was high (i.e., congruent trials; $t(6) = 1.18$; $p = 0.28$), likely due to reduced statistical power. We also did not observe significant coordinated reinstatement in the incongruent trials ($t(6) = 0.06$, $p = 0.949$).

In order to assess whether the coordination we found is specific to the encoding/retrieval time clusters we selected, we correlated ERS in the selected tROI of each region with ERS across the entire encoding/retrieval time space of the respective other region. We first correlated mean ERS in the HC tROI cluster with all different LTC encoding/retrieval time periods, using a sliding window of the same size as the previously determined LTC tROI (Fig. 5b). Conversely, we performed the same analysis over all encoding/retrieval time windows in HC, using the LTC tROI as a 'seed'. In both analyses, we included all available trials in the ERS$_{\text{same item,}}$

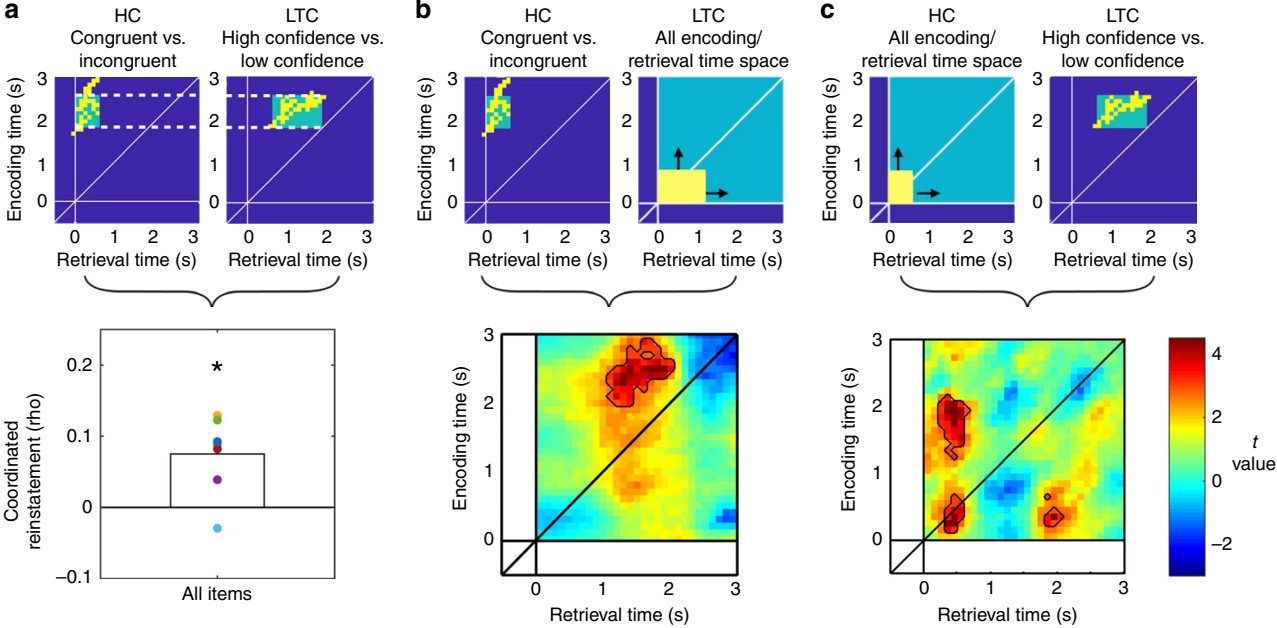

**Fig. 5** Coordination of representational reinstatement between HC and LTC. **a** Encoding time-matched temporal regions of interest (tROIs) were defined based on the significant clusters supporting reinstatement of item-context associations in the hippocampus and confidence effects in the lateral temporal cortex. Reinstatement values were correlated across trials in each patient. * indicates a p-value below 0.05 (one-sample t-test). **b** Correlation map between the hippocampal tROI cluster and all encoding-retrieval time pairs (0–3 s) in the lateral temporal cortex (cluster equally sized as in panel **a**, top right). **c** Correlation map between the LTC cluster and all encoding-retrieval time pairs in the HC. Bottom parts of panels **b** and **c** show T-maps in which correlations against zero at the group level were tested for the corresponding analysis. Areas in which values reached significance at $p_{(uncorrected)} < 0.05$ are outlined

all contexts condition. Correlations were maximal in similar time windows as the original LTC and HC tROI clusters (Fig. 5b, c). Although clusters in this analysis did not survive correction for multiple comparisons (all $p > 0.087$), these results nevertheless indicate a coordination mechanism which is temporally specific to the tROIs that have shown to be significant with respect to their representational format, i.e., encoding item or item-context information.

**Representational reinstatement and phase synchronization.** A possible mechanism that could underlie the coordination of reinstatement of different representational formats across brain regions is phase synchronization. Thus, in our final analysis, we investigated how representational reinstatement relates to the synchrony of the oscillatory phases between the hippocampus and the LTC. In order to obtain a measure of phase synchronization in each individual trial (and avoid inter-subject variability), the stability of phase differences between these two regions was calculated across time (Methods)[29]. We specifically focused, again, on the behaviorally relevant cluster of increased reinstatement observed in the confidence contrast in the LTC. Since this form of LTC reinstatement followed HC reinstatement during retrieval, we hypothesized LTC reinstatement to be preceded by an early LTC-HC phase synchronization, in particular in the gamma frequency range[30]. We calculated relative phase synchronization changes from baseline during the retrieval phase of our experiment in a 500 ms window after stimulus onset (i.e., during the time period of hippocampal reinstatement). For every frequency between 1 and 100 Hz, we correlated the magnitude of phase synchronization and LTC ERS across trials (using mean ERS in the LTC tROI cluster, Methods). Indeed, we observed significant correlations of LTC ERS with HC-LTC phase synchronization in frequencies between 77 and 83 Hz, with a mean peak at 79 Hz ($p_{(corr)} = 0.023$, Fig. 6a). The same relationship

could not be established with HC ERS. Indeed, no clusters survived correction for multiple comparisons when we compared HC-LTC phase synchronization between 0–500 ms with HC ERS in the HC tROI (all $p > 0.266$, Fig. 6b).

Taken together, these results demonstrate that representational reinstatement is coordinated across HC and LTC, and that HC-LTC gamma phase synchronization during the time period of HC reinstatement predicts LTC reinstatement.

## Discussion
We have addressed the fundamental question of how hippocampus and neocortex contribute to episodic memory by analyzing the distinct representational formats and coordination of hippocampal and neocortical reinstatement during context-dependent memory retrieval. We observed associative reinstatement of item-context information in the HC that precedes, and is correlated with, behaviorally relevant item reinstatement in the LTC. Previous work has provided evidence for reinstatement of oscillatory patterns representing item-specific information in neocortex[23,25,28] and item-context associations in the hippocampus[22]. Our results generalize these effects to a more ecologically valid task in which participants actively navigate within a virtual reality environment. More importantly, our findings demonstrate both a dissociation of representational reinstatement across time and space and their coordinated interaction across trials. In addition, our data shows a direct relationship between LTC reinstatement and HC-LTC phase synchronization in the gamma frequency range, providing direct evidence that this measure of dynamic coupling indeed reflects information transfer in the human brain.

Recent years have seen an upsurge in the application of multivariate analysis methods to human data in particular from neuroimaging. In the domain of memory research, this was accompanied by a shift in focus from the cognitive processes

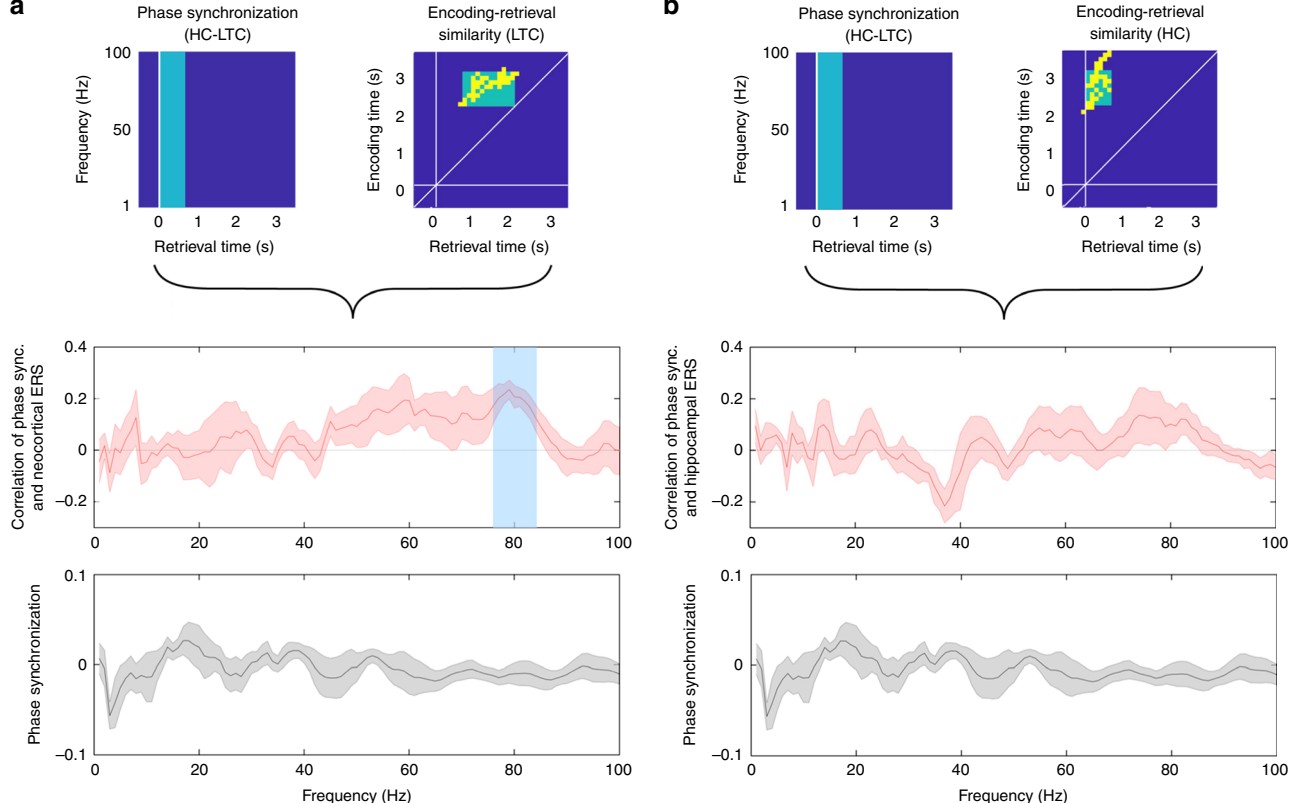

**Fig. 6** Gamma phase synchronization correlates with LTC reinstatement. **a** Frequency-specific (77–83 Hz) hippocampal-lateral temporal cortex phase synchronization significantly correlates with behaviorally relevant reinstatement in lateral temporal cortex (blue transparent box). **b** Hippocampal-lateral temporal cortex phase synchronization does not correlate with hippocampal reinstatement during the time period of increased hippocampal ERS. Curves and shaded areas in **a** and **b** show mean ± SEM of correlation of phase synchronization and LTC reinstatement (red), and mean ± SEM of baseline-corrected hippocampal-LTC early phase synchronization (0–500 ms at retrieval, black)

supporting memory functions in general to the representation of specific content[12,40]. Applying either pattern classification or representational similarity analysis, various fMRI studies have provided evidence for both hippocampal and neocortical representational reinstatement during retrieval (for a review see ref. [40]). However, several fundamental questions have so far not been answered: What are the representational formats employed, what is their relative timing and how are they coordinated? Here, we exploited the high temporal resolution and spatial specificity of intracranial EEG recordings to address these open issues.

With respect to timing, we observed that item-context associations were reinstated within the first 500 ms after the retrieval cue was presented. This result is in line with evidence indicating a flow of information from the hippocampus to the neocortex during human memory retrieval[11,13]. In contrast to other studies, we observed reinstatement of hippocampal information significantly earlier (e.g., around 1 s before ref. [22]) and immediately after stimulus onset. We believe this is due to the embedding of memory retrieval in an active navigation task in our study, where the presentation of a specific context during retrieval (i.e., navigation into a room) rapidly triggers reinstatement of items that had been learned in that context. In LTC, we observe that item representations were reinstated substantially later than hippocampal reinstatement, starting ~1 s post-stimulus, shortly before the response is given. These results are consistent with previous reports of reinstatement in LTC based on single-unit recordings[26]. However, to our knowledge, no previous study has simultaneously compared the reinstatement of different representational formats in hippocampus and neocortex.

In order to study the interaction between hippocampal and neocortical reinstatement, we applied a novel metric of representational coordination based on inter trial correlation of reinstatement in distinct spatio-temporal regions of interest (see also Supplementary Note 7). We found consistent positive correlations of hippocampal and neocortical reinstatement (Fig. 5). This coordination seems to be based on a generic mechanism because the effect was independent of contextual congruency. Importantly, however, representational coordination was not a general phenomenon across all encoding/retrieval time periods but was confined to the specific time periods showing representational reinstatement in the two areas.

We also observed a relationship between LTC reinstatement and HC-LTC gamma phase synchronization. Given the temporal offset of this synchronization pattern, it appears that phase-based functional connectivity between the two regions supports the coordinated reactivation of encoding-related representations. Indeed, our data suggests a dissociation between neural processes coordinating the interaction between areas and those supporting the representation of their respective specific content: while we observed that HC-LTC phase synchronization in the high gamma range (between 77 and 83 Hz) was linked to LTC reinstatement, item-specific representations in LTC were supported by power effects across a broad frequency range (Fig. 4b). The distinct temporal organization of HC-LTC phase synchronization and LTC reinstatement suggests that part of the information reinstated early in the hippocampus is transferred to LTC facilitating item-specific retrieval.

Our results in the coordinated reinstatement analysis and the phase synchronization analysis are consistent with theoretical

notions of hippocampal-neocortical interactions supporting episodic memory[1,3,4,6,11,12,41]. In the future, it will be important to investigate the directionality of these interactions and relate them to the representation of specific content at the single cell level (ref. [26], see ref. [20] for a review). Given the differences in the timings of reinstatement we observe relative to previous literature, and the fact that action and active learning promote hippocampal-neocortical interactions[41], we also highlight the need to further explore the specific role of active learning in modulating hippocampal and neocortical reinstatement.

The results presented here provide novel insights into the neurophysiological patterns that support representations in hippocampus and LTC during encoding and retrieval of contextual memories. Interestingly, while LTC reinstatement depended on information carried across a wide range of frequencies, hippocampal reinstatement was mostly driven by low-frequency oscillations. This contrasts with previous iEEG studies in which either broadband[22,42] or high-frequency oscillations[23,25] were reported to encode stimulus-specific information in the hippocampus (for a review, see ref. [20]). Given the active navigation task we deployed, one may hypothesize that the band-specific effects observed here are related to the tight coupling of hippocampal delta and theta oscillations with spatial navigation in humans (e.g., refs. [43–45]) which has been proposed to be a fundamental feature of hippocampus-dependent memory functions.

Our findings demonstrate a strong link between the reliability of stimulus-specific representations in LTC and confidence judgments. On one hand we observed a substantial temporal overlap between confidence-related and item-specific clusters in LTC, on the other we encountered a significant relationship between reinstatement and confidence in the clusters that were identified in the item-specific contrast. By contrast, hippocampal reinstatement was not related to retrieval confidence, in apparent contradiction to previous studies using single-units[39,46] and fMRI[47,48]. However, other findings suggest that during retrieval hippocampal activity depends mainly on the actual presentation of an item rather than on subjective confidence[49,50]. On the other hand, our multi-electrode analysis in the parietal lobe revealed a significant effect of confidence in reinstatement. This finding is consistent with the well-known role of the posterior parietal cortex in the processing of subjective memory confidence[39,51].

While further work is needed to clarify how subjective memory is represented in the brain, our data obtained in an active paradigm suggest a framework where hippocampal reinstatement of item-context associations is unrelated to declared retrieval confidence, while confidence-related signals are expressed in cortical regions. Please note that in the confidence analysis, we included all trials irrespective of memory performance (i.e., both hits and false alarms). Even though this analysis is less conventional, it has been employed in previous studies (e.g., refs. [38,39]) where it has been considered to reflect "subjective memory confidence". Indeed, confidence judgments have been argued to be an integral part of memory[38,39,46] and are typically included in standard tests of recognition memory[52].

Hippocampal reinstatement could not be explained by room-specific activity itself (Fig. 2c). Previous studies using intracranial EEG indicate that spatial locations can be decoded from population-level activity in humans, including the hippocampus[23] – conceptually similar to findings of location-specific local field potentials in rodents[53], albeit on a different spatial scale. By contrast, it is still a disputed question whether distributed patterns of BOLD responses allow for decoding of spatial positions[54,55] or not[56]; in general, the sparse and distributed nature of hippocampal representations[57,58] and the apparent lack of a topological organization[59] complicate decoding from population-level activity.

Even though it is a common practice to aggregate data collected from different hemispheres to increase statistical power in intracranial EEG studies, we compared reinstatement patterns in patients with electrodes in the left and right hemisphere. No difference was observed in the hippocampus; in LTC, the left but not the right hemisphere group showed an effect. A direct comparison did not reveal any significant hemispheric differences. The lack of pronounced lateralization effects may be explained by the contribution of medial temporal lobe structures of both hemispheres to spatial navigation and memory[60,61]. In this context, it should be noted that the prevalence of atypical language lateralization is higher in epilepsy patients than in the general population[62], which may obscure possible lateralization effects if language lateralization is not explicitly assessed. This is because in patients with atypical language lateralization, material-dependent memory functions often shift their hemispheric distribution as well[63].

We note that although we cannot completely rule out a possible influence of perceptual similarity between encoding and retrieval on reinstatement, such an effect cannot explain the patterns of reinstatement we observe. First, if perceptual similarity was affecting the results, it should have been identified in the same vs. different room contrast, but this is not what we observed in either hippocampus or LTC. Moreover, we found a strikingly different expression of item and contextual reinstatement in these brain regions, while a pure perceptual account would predict an identical response pattern. Finally, if the results were solely driven by perceptual similarity, there should be no difference in the relationship between reinstatement and reported confidence. However, as discussed above, we observed a relationship between LTC reinstatement and subjective confidence, which seems to be related to the reliability of stimulus-specific representations. Thus, it is highly unlikely that reinstatement effects can be explained by virtue of a perceptual overlap between encoding and retrieval. Nevertheless, given the relatively low number of error trials in our experiment, the exact contribution of perceptual factors cannot be fully quantified. Hence, in follow-up experiments more challenging memory tasks should be considered.

In summary, our results demonstrate a dissociation of representational reinstatement between hippocampus and lateral temporal cortex in terms of representational features, time during retrieval, and the contributing frequencies. In addition, we show that reinstatement is coordinated and linked to gamma phase synchronization across these brain regions. Taken together, we provide a novel and mechanistic explanation of how content-specific representations manifested in distinct oscillatory signatures of neuronal activity are orchestrated to support episodic memory. In this view, the hippocampus is multiplexing item-context information together with retrieval cues to re-establish a hippocampal-cortical memory network.

## Methods

**Participants**. Eleven epilepsy patients (6 males, 23–49 years) who had been surgically implanted with depth electrodes as part of their diagnostic assessment of surgical treatment for medically refractory epilepsy participated in our experiment. Some of them had to be excluded from specific analyses. For instance, all subjects with hippocampal seizure onset zone were excluded from analyses involving the hippocampus. On the other hand, all 11 subjects were included in the LTC contrasts, but we verified the results also in the group of $n = 8$ subjects without hippocampal epilepsy. Patient's demographic and clinical data is presented in Supplementary Table 1. The study was approved by the local ethics committee, "Clinical Research Ethical Committee (CEIC) Parc de Salut Mar" (Barcelona, Spain). All patients provided written informed consent to participate in the experiment.

**Task description**. Patients were asked to navigate a virtual maze comprising a central and four satellite rooms. Each room had a unique visual texture on its walls

(i.e., concrete, stone, brick, and wood). In each trial, patients were presented a texture and asked to navigate to the corresponding room. Each room contained a wall matrix of 5 x 4 = 20 images. During encoding, once they arrived in each room, they were presented with one specific image (presentation time, 1 s) and were asked to try to memorize it. Subjects encoded a total of 40 images. During retrieval, patients were asked to indicate whether they had seen the image during encoding, using a 6-points confidence scale — from 1 (sure unfamiliar) to 6 (sure familiar[52]). Participants saw all 40 encoding items again, randomly intermixed with 40 new items. Half of the old items were shown in the same room and at the same position on the wall matrix as during encoding (congruent items), while the other half were shown in a different room and position in the wall matrix (incongruent items). Note that in the incongruent condition, both the room and the wall position differed between encoding and retrieval for a given item. On average, a number of 5 trials were encoded in one room and retrieved in the same room but at a different position in the wall matrix, and those were excluded from the incongruent condition. Participants were not instructed to remember the spatial position where the items were presented, but only the items (i.e., encoding of rooms was incidental).

All items were extracted from a publicly available dataset[64]. From all objects in that dataset, 160 items were selected from different semantic categories (e.g., animals, fruits, buildings, tools). For each subject separately, a subset of 80 images was randomly selected from this pool of 160 images, and assigned (again randomly) to the different conditions. In total, 20 images were assigned to the congruent condition and 20 to the incongruent condition.

Before the start of the experiment, subjects performed a training session in which they could familiarize themselves with the maze and with using the joystick. The task in the training session was to reach 10 rooms one after the other (without encoding any items). The starting position for the first trial in each block was set to the central room for all participants. The sequence of rooms to visit was randomized. Patients performed the task on a 17" portable computer while sitting in their hospital bed. The VR application was created using the Unity3D game engine (Unity Technologies, San Francisco, CA, USA).

**Behavioral analysis**. The Receiver Operating Characteristic (ROC) curves were constructed by calculating the ratio of hits (old items correctly identified as old) versus false alarms (new items incorrectly identified as old) under different levels of confidence[52]. Note that the Area Under the Curve (AUC) illustrates the overall performance in the task (i.e., it takes into account performance for old and new items). We specifically addressed differences in recognition performance for congruent and incongruent trials by calculating recognition accuracy in each condition, which was defined as the number of trials correctly identified as old divided by the total number of trials in the correspondent condition.

Given that participants responded in general with high confidence (1 and 6 responses), and in order to allow for a comparison between conditions with equivalent numbers of trials, we pooled trials with responses 2-3-4-5 and labeled them as low confidence trials. Results in the behavioral plot (Supplementary Fig. 2C) and the intracranial EEG analysis (Figs. 3 and 4) use this definition of high and low confidence.

Response time at retrieval was quantified for each subject as the median time elapsed from stimulus onset until response. Mean response times across subjects was 1.69 s (S.E.M: 0.53 s); see plot in Fig. 3e.

**Electrophysiological recordings**. Recordings were performed using a standard clinical EEG system (XLTEK, subsidiary of Natus Medical) with 500 Hz sampling rate. A unilateral implantation was performed in all patients, using 7 to 10 intra-cerebral electrodes (Dixi Médical, Besançon, France; diameter: 0.8 mm; 5 to 15 contacts, 2 mm long, 1.5 mm apart) that were stereotactically inserted using robotic guidance (ROSA, Medtech Surgical, Inc, Montpellier, France).

**Electrode selection**. Electrodes were stereotactically implanted by our clinical team at the Hospital Del Mar (Barcelona, Spain). Targeted regions varied across patients for clinical reasons, but in all patients included the anterior hippocampus in left ($n = 7$) or right ($n = 4$) hemispheres. We selected only one electrode contact in the hippocampus and one in the LTC of each patient, in line with previous studies[22] (see also "temporal RSA" approach in ref. [25]). Our patients never had more than one electrode targeting the anterior hippocampus. From this electrode, we chose the most distal hippocampal contact and the contact that was located most centrally in gray matter in the lateral temporal cortex. We used a bipolar reference. Across all our patients, we selected a contact from the same electrode as the hippocampal contact in order to minimize variance in spatial location of the LTC contact (note that HC and LTC contacts correspond to the same hemisphere in all our subjects). We decided to select these two areas because of the well-documented relevance of the anterior hippocampus for relational memory (e.g., ref. [65]) and more specifically, based on the finding of increased reinstatement for source as compared to item memory in the anterior hippocampus[22]. The LTC has been previously linked to recognition memory and has been investigated in similar iEEG setups[26]. However, no previous study directly compared reinstatement in hippocampus and neocortex.

Sometimes more than one contact reached the hippocampus or was located in LTC grey matter (in the anterior and posterior regions of the HC, for instance, or

on two sides of the same gyrus in the LTC). The presence of specific contacts within the hippocampus and LTC was confirmed via careful examination of the MRI and CT scans. Electrode locations in native space were converted to MNI coordinates using 3D slicer (www.slicer.org[66]) and BrainX[3] (see ref. [67]), following the method described in ref. [68] (a full list of electrodes used and their MNI coordinates is presented in Supplementary Data 1). After co-registering pre- and post-electrode placement using MR images and CT whole-brain volumes, we could confirm 15 contacts located in the anterior hippocampus and 10 contacts in the posterior hippocampus (excluding patients with hippocampal epilepsy).

**Time-frequency analysis**. We band-pass filtered the signal at the selected electrodes from 1 to 200 Hz using a two-way, zero phase-lag, finite impulse response filter to prevent phase distortion (eegfiltnew.m function in EEGLAB toolbox[69]). Before decomposing the signal, we divided the data into 6-second epochs centered around stimulus onset in the encoding and retrieval phase of the experiment. We chose a long window to later remove edge artifacts occurring during wavelet decomposition. Using the FieldTrip toolbox[70], we decomposed the signal using complex Morlet wavelets with a variable number of cycles, i.e., linearly increasing between 3 cycles (at 1 Hz) and 6 cycles (at 29 Hz) in 29 steps for the low-frequency range, and from 6 cycles (at 30 Hz) to 12 cycles (at 100 Hz) in 15 steps. The resulting time-series of frequency-specific power were then decibel transformed by taking as a reference the activity from a baseline period of −500 milliseconds until stimulus onset[22]. We then visually inspected all raw signal and spectrogram epochs (at encoding and retrieval) for each subject independently and removed noisy epochs. Number of trials removed varied depending on the quality of the signal. In total, the number of trials included in the item-specific analysis was 34.71 ± 5.9 (mean ± standard deviation). In the congruent condition, of a maximum of 20 encoding retrieval time-pairs, 17.28 ± 3.2 were included. In the incongruent condition 13.42 ± 2.8 were included. In the high confidence condition, a total number of 25.4 ± 11.08 encoding-retrieval time-pairs was included, and 10.3 ± 8.6 in the low confidence condition.

**Single-electrode (local) reinstatement analysis**. We quantified the similarity of neural representations during encoding and retrieval by comparing epochs of brain activity at HC (anterior and posterior) and LTC electrodes via representational similarity analysis (RSA)[19]. As in previous studies[22,25], we calculated Spearman's correlations of broadband oscillatory patterns of activity across time, resulting in a measure of ranked similarity between two encoding and retrieval time windows at the same electrode. To assess the reinstatement of activity in specific trials, we first defined a 500 ms time window in which we included the time courses of 1 to 100 Hz power (1 Hz steps from 1–29 Hz, 5-Hz steps from 30–100 Hz) relative to a 500 milliseconds pre-stimulus baseline window. Given the sampling rate of the data (500 Hz), a representational pattern consisted of 44×250 values which were concatenated into a one-dimensional vector for correlation analysis. We calculated correlations between encoding and retrieval time windows proceeding in time steps of 100 ms. Please note that in all plots of encoding-retrieval similarity (ERS), correlations corresponding to each 500 milliseconds window were assigned to the time point at the beginning of the respective window (e.g., a time bin corresponding to activity from 0 to 500 milliseconds was assigned to 0). Similar to refs. [22] and [23], we compared not only the activity during corresponding encoding and retrieval time windows, but correlated activity during all encoding time windows with activity during all retrieval time windows. This resulted in an encoding x retrieval reinstatement map for each trial including non-lagged (on-diagonal) and lagged (off-diagonal) correlations (Fig. 1d). The obtained reinstatement maps were subsequently Fisher z-transformed for statistical analysis and contrasted via paired t-tests across conditions of interest.

**Multi-electrode (global) reinstatement analysis**. In addition to our analyses based on single electrode data, we performed additional analyses in which we included activity across all anterior and posterior hippocampal contacts ($n = 25$); across multiple contacts in the lateral temporal lobe ($n = 117$); and across multiple contacts in the parietal lobe ($n = 19$; see also Supplementary Data 1). In all of these novel analyses, we built representational patterns based on distributed broadband oscillatory power across multiple contacts, i.e., from all electrodes available for each subject in a given region of interest. Patterns of each contact were built using the same parameters for time and frequency resolution as explained in the local RSA analysis, but were then concatenated into one "global" feature vector before performing the similarity comparisons.

**Contrasts**. We explored encoding-retrieval similarity (ERS) for trials in congruent and incongruent spatial context conditions. Congruent trials were defined as those for which one item was presented in the same room and position on the wall matrix at encoding and retrieval. Incongruent trials were defined as those in which an item was presented in one room at encoding and in a different room and position in the wall matrix at retrieval.

In order to study item-specific reinstatement, we first calculated ERS for all 40 old trials. To assess whether the information reinstated in these trials was item-specific, we created surrogate ERS values by correlating the activity during encoding of one item with the activity during retrieval of a different item. We

calculated ERS for all "same item" and "different item" pairs, averaged across trials in each subject and then calculated statistical tests between the different conditions at the group level.

To assess whether reinstatement was only due to contextual information irrespective of items, we calculated ERS between items that were encoded and retrieved in the same room (irrespective of item or position in the wall matrix). We then created surrogate "different room" comparisons, by pairing items encoded in one room and retrieved in a different room (again regardless of item identity). ERS for same-room and different-room correlations were each averaged across trials in individual subjects and compared at the group level.

To control for differences in ERS due to the different numbers of trials in the surrogate conditions (e.g., "different room" as compared to "same room", or "different items" as compared to "same items"), we performed the same analysis by randomly selecting trials from the conditions with more samples before performing statistical testing. We observed the same pattern of results in this control analysis.

In order to assess the behavioral relevance of reinstatement in the HC and the LTC, we compared reinstatement for high versus low confidence trials. We included all trials in this comparison (congruent + incongruent, irrespective of context). Three out of eleven participants responded in all trials with high confidence and were therefore excluded from this analysis. We also excluded these participants from all subsequent analyses that were based on the significant cluster observed in this comparison.

Note that in all our main contrasts (Figs. 1 and 2), we included all trials available for each condition, irrespective of memory performance. In separate analysis, we specifically calculated ERS for correct and incorrect trials (Supplementary Figs. 9–11).

We also directly compared reinstatement in hits versus miss trials, in hits versus correct rejections, and in high confidence hits versus high confidence correct rejections. In the hits versus misses analysis, we compared old items that were correctly identified as old versus old items that were incorrectly classified as new. In the hits versus correct rejection analysis, the correct rejection condition was built by comparing activity of all items presented at encoding with all novel items presented at retrieval that were correctly identified as novel. The same was done in the high confidence hits versus high confidence correction rejection analysis, where we only included high confidence trials (i.e., responses 1 or 6).

We split the correct rejection condition in congruent and incongruent trials to assess the interaction between hits versus correct rejections and congruent versus incongruent. The correct rejection congruent condition was built by correlating all items presented in one room at encoding with all correctly identified novel items presented in the same room at retrieval. The correct rejection incongruent condition was built by calculating ERS between all items presented in one room at encoding with activity from all correctly identified novel items in a different room at retrieval.

To test for significant differences in the conditions of interest (e.g., congruent vs. incongruent, same item vs. different item, same room vs. different room, high confidence vs. low confidence), we performed a paired $t$-test at each encoding-retrieval time pair, in which we included the mean ERS value for each subject in each condition. Correction for multiple comparisons was performed using cluster statistics (see below).

Please note that we do not report differences in mean reinstatement values across conditions in our bar plots (Figs. 2d, 3b, 3f). These figures and analyses present additional information including statistical comparisons against zero at the group level and the individual means in the clusters of interest for each condition.

**Frequency-specific analysis and jackknife procedure**. In the frequency-specific ERS analysis, we constructed "single frequency" representational patterns in three frequency bands: delta (1–3 Hz), theta (4–8 Hz) and delta/theta (1–8 Hz). Delta and theta were combined because of the ongoing discussion about the human correlate of rodent theta oscillations, in particular during spatial navigation[44,71,72].

Indeed, theta oscillations comprise a functionally relevant frequency band which has been linked in the animal literature to important cognitive processes such as active learning, memory encoding, and spatial navigation (e.g., refs. [73,74]). A similar function of this frequency band has been hypothesized to exist in humans, albeit at slightly slower frequencies (1–3 Hz, or delta; see refs. [71,72,75]). Based on these results, it has been suggested that the frequency of human theta oscillations is lower than in rodents[76], possibly due to the larger anatomical extent of neural assemblies in humans as compared to rodents[77]. On the other hand, a recent study compared virtual and actual physical navigation and described theta oscillations at a higher frequency during real world as compared to virtual navigation, even though theta oscillations at a lower frequency occurred as well[44]. Thus, similar to rodents (e.g., ref. [78]), there may be multiple theta generators in the human hippocampal formation that have different frequency profiles and distinct – sometimes even opposing (e.g., ref. [79]) – functional roles. In the current study, we used a parsimonious approach and analyzed the contribution of activity across an extended frequency range including both conventional delta and theta oscillations (combined band: 1–8 Hz).

To quantify the contribution of delta, theta and delta/theta to reinstatement we implemented a jackknife procedure. We constructed representational feature vectors by concatenating the time series of oscillatory power of the

corresponding frequencies only (no average was made across frequencies). We used the same parameters as in the broadband ERS analysis regarding window size (500 ms), overlapping (80% overlap) and measure of similarity (Spearman's rho). We selected the specific clusters resulting from the HC congruent/incongruent contrast and the LTC same vs. different item and high vs. low confidence comparisons (only subjects who provided low confidence responses and without hippocampal epilepsy were included in this analysis). We calculated mean difference ERS in these clusters (i.e., congruent minus incongruent, high confidence minus low confidence and same item minus different item), when including all frequencies, and subtracted from these values those calculated after removing the information contained in the specific frequency band of interest.

**Coordinated reinstatement analysis**. To assess whether reinstatement in hippocampus and lateral temporal cortex was coordinated, we first defined two temporal regions of interest (tROIs) in the reinstatement maps of the hippocampus and the LTC. We selected the respective conditions in which we had observed significant intra-regional reinstatement effects – i.e., the congruent-incongruent contrast for the HC and the confidence contrast for the LTC, respectively. We created a rectangular mask over the reinstatement map of those contrasts and defined its limits by matching the encoding times of the corresponding clusters. We only included those encoding time bins in which at least two significant encoding-retrieval time pairs were observed in the clusters of the hippocampus or the LTC. This resulted in a window from 2.5–3.1 s at encoding for both contrasts and from 0.6–1.1 s at retrieval in the hippocampus, and 1.3–2.3 s at retrieval in the LTC (Fig. 5a). For each subject, we calculated the mean ERS between encoding of one item and retrieval of the same item in these rectangular tROIs at each contact and in every trial. Please note that this was performed across trials within individual participants, resulting for each subject and each trial in two average ERS values, one for each electrode and its respective tROI. We used Spearman's rho to assess the correlation between the two sets of values, resulting in one "correlation of correlations" value per subject across trials. We then Fisher $z$-transformed the rho scores and performed a $t$-test against zero at the group level. The same method was followed in the coordinated reinstatement analysis between the hippocampus and the posterior parietal cortex presented in Supplementary Note 10 and Supplementary Fig. 20.

To address the temporal specificity of the coordination of hippocampal and LTC reinstatement, we performed the same analysis by correlating reinstatement in each of our two tROIs with reinstatement in all possible encoding-retrieval time periods in the respective other contact. We ran two separate analyses. We first used the hippocampal tROI as a "seed" and correlated averaged ERS in this tROI with that of a sliding time window that covered the whole time-space in the LTC data. Conversely, we used the LTC tROI as a seed, and correlated mean ERS in this tROI with that of a sliding time window that covered the whole encoding-retrieval time space in the hippocampal ERS map. The size of the sliding tROIs was taken from the previous analysis (Fig. 5a). We color-coded the results over a new encoding-retrieval time map with $t$-values obtained from the comparison of Fisher $z$-transformed rho values against zero. If coordination is specific to the tROIs where the representational features of each channel are maximally expressed, this analysis would reflect increased correlations in those same locations in time (Fig. 5a). Note that the values obtained from the correlation of mean ERS in the sliding tROIs with the correspondent tROI of the other region were plotted at the center of the sliding tROI in the coordinated reinstatement map at encoding and retrieval. Given the size of the sliding windows, we did not include in the plots activity before the onset of the stimuli and restricted the time-space to 3 s instead of 3.5 as in the previous plots. To assess the significance of this cluster we performed cluster statistics as described in the multiple comparisons correction section. We conducted this analysis separately for all trials, corresponding to the (item same, all contexts) condition and for the subset of congruent and incongruent trials.

**Analysis of phase synchronization**. We calculated single-trial values of phase synchronization via a temporal version of the "Phase Locking Value" (PLV) to investigate synchronization between the hippocampus and the LTC according to the following formula:

$$\mathrm{PLV}f = \left| n^{-1} \sum_{t=1}^{n} e^{i(\varphi_{xt} - \varphi_{yt})} \right| \qquad (1)$$

In which $n$ is the number of time points and $\varphi_{xt}$ and $\varphi_{yt}$ are phase angles from electrodes $x$ and $y$ at frequency $f$[29].

In order to analyze phase synchronization at a higher frequency resolution, we extracted frequencies in steps of 1 Hz for this analysis. We first defined a temporal region of interest from 0 to 500 ms after stimulus onset at retrieval, and calculated for each trial the phase locking value in this tROI by subtracting PLV during a baseline window between 500 ms prior to stimulus onset until 0. We subsequently correlated baseline-corrected single-trial PLV values and mean LTC ERS in the behaviorally relevant tROI extracted from the confidence contrast. Please note that as in the coordinated reinstatement analysis, this correlation analysis was conducted across individual trials for each subject independently, which in turn required us to analyze phase synchrony across time in our specific time window

(0–500 ms). We compared the Fisher $z$-transformed Rho values against zero at the group level to assess statistical significance.

**Multiple comparisons correction.** Cluster-based permutation statistics were used to correct for multiple comparisons in all ERS contrasts and the coordinated reinstatement analysis. For the ERS contrast analyses, we created a null distribution of ERS values by permuting the labels of the trials for each subject independently 1000 times. We extracted for each permutation the sum of the $t$-values of the largest cluster, and only considered significant those contiguous encoding-retrieval time-pairs in the non-shuffled data whose summed $t$-values exceeded the summed $t$-value of 95% of the distribution of surrogate clusters (corresponding to a corrected $p < 0.05$)[80]. Note that the same pattern of results was observed when, instead of shuffling trial labels, we shuffled the condition averages (i.e., after averaging across the condition-specific single-trial correlations at the level of individual participants). In addition, to avoid any bias in the contrasts with unbalanced trial numbers, we performed the same permutation procedure by randomly selecting a subset of trials from the condition with more trials to match the number of trials in the condition with fewer trials. All significant clusters in the different contrasts (LTC: same item vs. different items, high confidence vs. low confidence; HC: congruent trials vs. incongruent trials) also survived when applying this procedure.

In the coordinated reinstatement analysis, we shuffled between trials selected at the HC contact and trials at the LTC contact, leaving encoding-retrieval assignments at each contact unaffected. For example, we extracted ERS between encoding of item #6 and retrieval of item #6 in HC and between encoding of item #34 and retrieval of item #34 in LTC. We repeated this procedure 1000 times, extracting for each iteration the sum of $t$-values from the largest cluster of correlations significantly greater than zero. This resulted in a distribution of $t$-values under the null hypothesis; then, we only considered significant those clusters whose summed $t$-values were above the 95th percentile of the distribution of surrogate clusters.

## Data availability

The data that support the findings of this study are available on reasonable request from the corresponding author (P.V.).

## Code availability

Standard software packages (EEGLAB, Fieldtrip) were used for processing the iEEG data in addition to custom Matlab scripts. Custom-written code is available upon reasonable request from the corresponding author (P.V.).

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

## Acknowledgements

We thank Riccardo Zucca, Juergen Fell, Marie-Christin Fellner and Lukas Kunz for helpful comments on an earlier version of the manuscript; Sytse Wierenga and Antoni Gurguí for their help in the production of figures, and Carmen Pérez for the conversion of electrodes positions to MNI space. The research leading to these results has received funding from the ERC grant agreement n° 341196 (CDAC). N.A. received funding by the Deutsche Forschungsgemeinschaft (DFG, German Research Foundation) – Projektnummer 316803389 – SFB 1280 as well as via Projektnummer 122679504 – SFB 874. M.S. received funding by Spanish National Project INSOCO-DPI2016-80116-P.

## Author contributions

Conceptualization, D.P., A.D., M.S., and P.V.; Methodology, D.P., H.Z., N.A. and P.V.; Data Collection, D.P., Formal Analysis: D.P., P.V. and N.A.; Writing – Original Draft, D.P.; Writing – Review & Editing, P.V. and N.A.; Funding Acquisition, P.V.; Resources, R.R. and A.P.; Supervision, P.V. and N.A.

## Additional information

**Competing interests:** The authors declare no competing interests.

