## [Peer Review File · Nature Communications]

Reviewers' comments:

Reviewer #1 (Remarks to the Author):

This study examines an important question related to the neural mechanisms that underlie episodic memory formation, namely the interaction between medial temporal lobe structures such as the hippocampus and the lateral temporal cortex. Current theories suggest that the hippocampus is responsible for storing associations between an item and its context, whereas the cortex stores detailed information about individual items. Here the authors set out to empirically test whether these representations are indeed dissociated, and whether these structures interact with one another during successful retrieval. They use intracranial EEG (iEEG) recordings to construct measures of reinstatement between encoding and retrieval, and then a number of sophisticated analyses to demonstrate how these measures may support the hypotheses related to the hippocampus and the cortex that they test.

This is an important question, and providing mechanistic insight into the process of episodic memory retrieval would be a valuable contribution to the field. Certainly, theories suggesting this dissociation in representation enjoy computational modeling support and some experimental support, but further direct empiric support from human data would be a welcome addition. As the authors note, previous studies have demonstrated that these various aspects of memory have been demonstrated, but here the authors provide the novel contribution of examining simultaneous representations in the hippocampus and cortex to probe their interaction. The study is well conducted and well written, and some of the conclusions appear to be supported, although there are a few issues that I feel should be addressed to strengthen the quality of the manuscript and the conclusions that are inferred.

One major limitation is that no comparisons in this study are made between correct and incorrect trials. The comparisons instead are between congruent and incongruent trials, and between similar and different items and contexts. This makes any interpretations of reinstatement difficult, since it is not clear whether such reinstatement is related to successful memory retrieval. Moreover, there is a behavioral limitation in that there were no actual differences in performance between congruent and incongruent trials. All of this raises the question of interpretation, and whether the differences in ERS here are related to memory or simply to perceptual processing. The authors note this limitation, and the fact that the majority of trials are correct, making such analyses difficult. They should clarify whether the analyses provided are only on the correct trials. In addition, they should at least provide the data and show the comparisons with the limited number of incorrect trials that were captured.

In lieu of this comparison, the authors instead provide an analysis comparing high confidence versus low confidence trials. This is meant to be a surrogate for memory, yet the authors did not find a difference in the HC with their comparison. However, it is not clear if they are lumping all high confidence trials together (the 1s and 6s), or just the high confidence recognition trials. It seems like they should only look at the hits in that analysis, since the hits would specifically require a similar pattern of activity during encoding and retrieval.

An interesting comparison could also be between the high confidence hits and the high confidence rejections (or even all of the hits versus all of the rejections). In this case you would expect to see a clear difference in reinstatement since no previous pattern of activity should be reinstated (since this is a novel item). This comparison should show that the LTC can distinguish these conditions, particularly if the claim is that the LTC captures item specific ERS. But its possible the HC would as well, which would also suggest some item specificity. This could be investigated by looking at same versus different contexts, etc, when comparing hits versus rejections.

The ERS in the HC appears quite early during retrieval (0 to 0.5s). This seems almost too early,

and raises the concern of whether this is a visual response (e.g., the same visual response to the item and/or context upon seeing the item during encoding and retrieval). Again, this would be addressed if there were a proper comparison between correct and incorrect trials. Interestingly, in the HC the ERS does not distinguish same from different items or contexts, which may actually argue against a visual response. Nonetheless, an additional analysis could be performed in which the authors lock the data to the actual response. Presumably, any evidence of reinstatement should occur right before the subjects make their response, and would be more reassuring that this is related to the process of memory.

When looking at the interaction between HC ERS and LTC ERS, the authors perform a correlation between the two average values. But for this analysis, they examine the ERS in the LTC for same item, all contexts. If they are examining behaviorally relevant item specific reinstatement in the LTC, then they should correlate the difference in HC ERS between congruent and incongruent with the difference in LTC ERS between same and different items.

The analysis of phase synchrony is interesting, but seems quite complicated. They correlate levels of phase synchrony during different time windows with LTC ERS (which LTC ERS?). This is an indirect test. Why not directly examine phase synchrony between the two structures in the different conditions? Is there a difference between conditions, and if so, does then this difference correlate with the differences in ERS? In addition, why does the absence of a correlation between synchrony and HC ERS mean that there is no recurrent influence from LTC to HC. This does not address the direction of influence, and the timing alone of the respective ERS would suggest that this should not occur.

Methodologically, all of the analyses are based on examining the time series data from one electrode contact each in the HC and LTC in each patient. The ERS is calculated based on this time series. But how is this one electrode chosen in each patient? Surely some patients have more than one electrode placed in the hippocampus, for instance, which means they have more than one set of two 'deepest' contacts to choose from. Same with the LTC. What do the data look like for other electrode choices? Was there any consideration given the performing the analyses here using multiple contacts in the HC and the LTC, constructing ERS based on the time-space patterns of activity across multiple contacts?

Why were so many more incongruent trials removed manually because of noisy epochs than congruent trials (20->13 versus 20->17). This raises the concern as to whether there was some bias in the removal process. Perhaps an automated removal process would be more robust, and remove a more equal distribution of trials.

Reviewer #2 (Remarks to the Author):

The authors propose to investigate the theory of human episodic memory that proposes that detailed sensory representations of individual items rely on the neocortex whereas the hippocampus realizes an index to these cortical representations through item-context associations. This theory predicts information flow from neocortex to hippocampus during memory formation and the reverse direction during memory retrieval.

The authors study patients with intractable epilepsy implanted with intracranial EEG in the hippocampus and lateral temporal cortex for clinical purposes in 7 patients. They used one hippocampal and one lateral temporal cortex electrode from the same hemisphere in each patient. They excluded 2 patients because the seizure onset zone was in the unilaterally implanted hippocampus. However, some patients had electrodes in the left and others in the right hemisphere. Subjects performed a Virtual Reality active navigation task involving a recognition

memory test (congruent and incongruent conditions).

They calculated ERS (Encoding-Retrieval Similarity) between feature vectors and found significantly higher ERS in congruent versus incongruent trials. They did not find evidence for reinstatement of item-specific information in the hippocampus. In addition, they found that ERS values were significantly higher than zero in congruent trials but not incongruent trials.

In the lateral temporal cortex they found item specific ERS values with 2 time windows during encoding that are reinstated during a retrieval time window between 1.5-2.5 seconds. In contrast to findings in the hippocampus they did not observe reinstatement of item-context associations.

They also investigated the behavioral relevance of their findings and did not detect a difference in task performance although the participants recalled most items. They investigated high confidence versus low confidence. Higher confidence trials were associated with higher levels of reinstatement in the left temporal cortex and not the hippocampus. Based upon these findings the authors conclude that item representations in lateral temporal cortex and item-context associations in the hippocampus and also temporally with item-context (hippocampus) occurring earlier during retrieval than item reinstatement (lateral temporal cortex).

In the hippocampus the delta/theta bands were found to be significant in reinstatement of item-context associations. There was not item-specific frequency band in the left temporal cortex. As the authors point out, these findings are in contrast to prior ECoG studies in the hippocampus.

The authors also investigated a possible mechanism underlying coordination of reinstatement – phase synchronization. They observed a correlation in lateral temporal cortex ERS and Hippocampus-lateral temporal cortex phase synchronization from 70-82 Hz (gamma) but no correlation in the opposite direction (from lateral temporal cortex to hippocampus) suggesting the hippocampus coordinated lateral temporal cortex response.

Patients: all had unilateral implantations of stereotactic EEG with 7-10 electrodes with 5-15 contacts. In figure 3 the lateral temporal electrodes appear to be fairly closely localized in an axial plan but more detail is needed. Figure 1: From figure one it appears that there were 3 right sided and 4 left sided electrode contacts included.

The study was performed rigorously and methods are well described and logical. The findings are novel. However, the number of subjects included in the study are limited and the authors only present data from the electrodes in the lateral temporal cortex and anterior hippocampus. In addition, roughly half of the subjects have left implantations and half have right implantations. There is no discussion regarding the limitations of having a mix of right and left sampling and the authors do not present results of each hemisphere individually. Another major criticism related to the lack of control electrodes analyzed. Each subject had 5-15 electrode contacts implanted. I suggest that the authors do additional analysis in a non-lateral temporal lobe cortical region to ascertain whether the findings are specific to the lateral temporal lobe relationship with the anterior hippocampus. For instance, use a parietal cortical contact pair as a control. Finally, there is very little detail regarding the patient characteristics (e.g. age, seizures onset region on ECoG, gender, etc.) This should be included.

Minor point:

Line 171: lower- confidence should be low confidence

Reviewer #3 (Remarks to the Author):

In this paper, Pacheco and colleagues report an intracranial EEG experiment investigating memory reinstatement in hippocampus and lateral temporal cortex (LTC). They recorded simultaneously from both regions, allowing them to relate hippocampal reinstatement to LTC reinstatement effects, which were temporally related and correlated across trials. Interestingly, hippocampal effects were modulated by the match of both item and context information at encoding and retrieval, whereas LTC effects were modulated only by item match, consistent with models of hippocampal function that emphasize contextual binding processes.

In general, I thought the paper was timely, theoretically interesting, and important in its advances beyond the existing literature. The use of intracranial EEG allowed for more sophisticated evaluation of the temporal relationship between different aspects of reinstatement than has been previously documented (e.g., with fMRI or with EEG studies recording from a single region). Such hippocampal-neocortical interactions factor heavily in neurobiological models of memory. The paper was also well-written. I have relatively few comments.

The authors should elaborate further on their decision to group the delta and theta bands. It would seem warranted to report the results for delta and theta separately (in addition to together, which does seem motivated by the literature).

It is unclear what is meant by "In order to exclude that effects were driven by global and content-unspecific event-related effects during encoding..." (lines 235-238), or how the analysis controlled for that possibility.

The correlation of hippocampal and LTC reinstatement was conducted over trials for ERS_same-item, all-contexts. But the hippocampal effect was specific to the same-item, same-context comparison (vs same-item, different-context). Shouldn't the correlation be limited to these trials? How many correlations were computed, and were they corrected for multiple comparisons? This issue is mentioned in the Discussion (lines 339-340) but it should be unpacked further.

Figure 6 - I found it confusing that the black and red lines were displayed on the same plot, since they seem to refer to different types of measures.

Response: Thank you for giving us the opportunity to revise our manuscript. As you will see, we have responded to all concerns that were raised by the 3 Referees. As recommended by Referee 2, we have also increased the sample size, by recording data from 2 additional subjects and including data from the lateral temporal cortex of 2 further patients. Our sample size is now at n=11 (although some data cannot be used for all analyses), which is larger than in the majority of intracranial EEG papers published in high-impact journals. We could corroborate all of our main results with this larger sample.

Reviewers' comments:

Reviewer #1 (Remarks to the Author):

This study examines an important question related to the neural mechanisms that underlie episodic memory formation, namely the interaction between medial temporal lobe structures such as the hippocampus and the lateral temporal cortex. Current theories suggest that the hippocampus is responsible for storing associations between an item and its context, whereas the cortex stores detailed information about individual items. Here the authors set out to empirically test whether these representations are indeed dissociated, and whether these structures interact with one another during successful retrieval. They use intracranial EEG (iEEG) recordings to construct measures of reinstatement between encoding and retrieval, and then a number of sophisticated analyses to demonstrate how these measures may support the hypotheses related to the hippocampus and the cortex that they test.

This is an important question, and providing mechanistic insight into the process of episodic memory retrieval would be a valuable contribution to the field. Certainly, theories suggesting this dissociation in representation enjoy computational modeling support and some experimental support, but further direct empiric support from human data would be a welcome addition. As the authors note, previous studies have demonstrated that these various aspects of memory have been demonstrated, but here the authors provide the novel contribution of examining simultaneous representations in the hippocampus and cortex to probe their interaction. The study is well conducted and well written, and some of the conclusions appear to be supported, although there are a few issues that I feel should be addressed to strengthen the quality of the manuscript and the conclusions that are inferred.

Response: We thank the referee for this overall positive assessment of our manuscript and for her/his detailed comments and suggestions. We below present our responses and indicate where new information can be found in the original manuscript.

One major limitation is that no comparisons in this study are made between correct and incorrect trials. The comparisons instead are between congruent and incongruent trials, and between similar and different items and contexts. This makes any interpretations of reinstatement difficult, since it is not clear whether such reinstatement is related to successful memory retrieval. All of this raises the question of interpretation, and whether the differences in ERS here are related to memory or simply to perceptual processing. The authors note this limitation, and the fact that the majority of trials are correct, making such analyses difficult. They should clarify whether the analyses provided are only on the correct trials. In addition, they should at least provide the data and show the comparisons with the limited number of incorrect trials that were captured.

Response: We agree that this is a relevant point. The results we presented in the previous version of the manuscript were not only on correct trials but on all trials. This is now clearly stated on I. 1003-1006, p. 46-47 of the revised manuscript.

We agree that it is important to show that the same results can also be obtained when only analyzing correct trials. In the revised version, we added analyses in which we only included trials with correct responses. In these analyses, we could replicate our main findings: In the hippocampus, we found a significant cluster at a very similar point in time (please note that this analysis relies on a larger sample of 8 patients, because we increased the sample size following a comment of Referee 2):

Hippocampus correct trials (n = 8)

Supplementary Figure S9: Congruency effects in correct trials in the hippocampus.

In the revised version, this is now presented as Supplementary Figure S9.

When we restricted the LTC analysis to correct trials, we also found two significant clusters in similar time periods as the original clusters (now with sample sizes of either $n=8$ and $n=11$ subjects, depending on inclusion criteria; see below):

Lateral temporal cortex correct trials (n = 8)

Lateral temporal cortex correct trials (n = 11)

Supplementary Figure S10: Reinstatement of item-specific information in the lateral temporal cortex in correct trials. (A,B): Reinstatement effects in $n=8$ patients without hippocampal epilepsy. (C,D): Reinstatement effects in the extended group of $n=11$ patients.

In the revised version, this is now presented as Supplementary Figure S10.

We also performed this analysis for the subset of incorrect trials, despite the low number of incorrect trials that were captured (only in 7 of 11 subjects; trial numbers: 3, 21, 16, 6, 4, 2, 2). No clusters were observed in either hippocampus (all $p > 0.272$) or LTC (all $p > 0.159$; see figure below).

Supplementary Figure S11: Lack of reinstatement of item-specific activity during incorrect trials in the hippocampus (A) and the lateral temporal cortex (B).

We also directly compared reinstatement in correct and incorrect trials. In the LTC we observed that, numerically, ERS was higher for correct vs. incorrect trials around the same temporal region of interest where we had observed a confidence effect before (even though no cluster survived correction for multiple comparisons; see figure below). When we restricted analysis to the cluster where a confidence effect had been observed, mean ERS for correct items was significantly greater than zero ($t(7) = 2.893$, $p = 0.027$; and $t(10) = 3.5897$, $p = 0.004$ in our extended group of $n=11$ subjects), while it did not differ from zero in the incorrect trials ($t(5) = -0.546$, $p = 0.608$; and $t(6) = -0.661$, $p = 0.533$ in the extended group).

Similarly, when we focused our analysis on the LTC reinstatement cluster ii (resulting from the orthogonal contrast of same vs. different items during encoding and retrieval) in a similar time period, we found that reinstatement values were significantly greater than zero for correct trials ($t(6) = 4.251$, $p = 0.005$; and $t(10) = 5.395$; $p = 0.0003$ in the extended group), but not for incorrect trials ($t(5) = -0.338$, $p = 0.748$; and $t(6) = -0.401$, $p = 0.701$ in the extended group).

Lateral temporal cortex correct versus incorrect trials (n=6)

Lateral temporal cortex correct versus incorrect trials (n=7)

Supplementary Figure S12: Direct comparison of reinstatement during correct and incorrect trials in the lateral temporal cortex. (A,B): Patients without hippocampal epilepsy. (C,D): Extended group of patients.

In the hippocampus, no differences were observed between correct and incorrect trials (all $p > 0.378$).

These results – in addition to our previous findings of confidence effects – suggest that item-specific reinstatement is indeed related to participants' memory rather than reflecting only perceptual processing.

We have included these results in the revised manuscript (Supplementary Note 3).

Taken together, results from the analyses on correct and incorrect trials suggest that reinstatement is linked to memory dependent performance. A role for perception seems unlikely given the marked differences in reinstatement between the correct and incorrect trials. Yet, we note in the discussion that "... given the relatively low number of error trials in our experiments, the exact contribution of perceptual factors cannot be fully quantified. Hence, in follow-up experiments more challenging memory tasks should be considered" (p. 18, l. 444-447).

In lieu of this comparison, the authors instead provide an analysis comparing high confidence versus low confidence trials. This is meant to be a surrogate for memory, yet the authors did not find a difference in the HC with their comparison. However, it is not clear if they are lumping all high confidence trials together (the 1s and 6s), or just the high confidence recognition trials.

Response: We apologize for the lack of clarity. In this analysis, we are grouping all high confidence trials together (i.e., the 1s and the 6s). Even though this analysis is less conventional, it has been employed in several previous studies (e.g., Rissman et al., *PNAS* 2011; Rutishauser et al., *Neuron* 2018) where it has been considered to reflect “subjective memory confidence”. Indeed, in addition to memory performance, recognition memory tests typically imply that subjects report their responses on a confidence scale. Confidence judgments have been argued to be an integral part of memory and linked to a hypothetical subjective memory signal (Rissman et al., *PNAS* 2011, Rutishauser et al., *Neuron* 2018). We have now elaborated on the rationale for this analysis in the Discussion section (p. 17, l. 405-410).

It seems like they should only look at the hits in that analysis, since the hits would specifically require a similar pattern of activity during encoding and retrieval. An interesting comparison could also be between the high confidence hits and the high confidence rejections (or even all of the hits versus all of the rejections). In this case you would expect to see a clear difference in reinstatement since no previous pattern of activity should be reinstated (since this is a novel item). This comparison should show that the LTC can distinguish these conditions, particularly if the claim is that the LTC captures item specific ERS. But its possible the HC would as well, which would also suggest some item specificity. This could be investigated by looking at same versus different contexts, etc, when comparing hits versus rejections.

Response: We thank the referee for this suggestion. In the revised manuscript, we now also compare hits versus correct rejections and high confidence hits versus high confidence correct rejections. For these analyses, we compared ERS of correctly remembered previously seen items (hits) with ERS of correctly identified novel items (correct rejections; see Methods, l. 1007-1015, p. 47). The same analysis was then also conducted by only focusing on the high confidence hits and high confidence correct rejections. These novel analyses (in the LTC) revealed increases in ERS at very similar positions in time as those observed in our main analyses.

In the hits versus correct rejection analysis, we observed a significant cluster of increased ERS ranging from 2 to 2.5 seconds in encoding time and from 1.3s to 2.2 seconds in retrieval time ($p = 0.040$). These time windows overlap substantially with cluster ii from our original analysis (see top row in the figure below).

In the high confidence hits versus high confidence correct rejection analysis, an even more extended cluster of reinstatement was observed, ranging from 1.5 to 3.5 seconds at retrieval and from 2 to 3.5 seconds at encoding ($p = 0.01$; see bottom row in the figure below).

In the revised manuscript, we have added these results as Supplementary Note 1 and Supplementary Figures S13 and S14. Taken together, these results confirm our original finding of item-specific and behaviorally relevant ERS increases in the LTC after ~1.5 seconds during retrieval.

Lateral Temporal Cortex (n = 11)

Hits versus Correct Rejections

High Confidence Hits versus High Confidence Correct Rejections

Supplementary Figure S13: Comparison of reinstatement during hits as compared to correct rejections in the lateral temporal cortex (extended group of patients). (A) Reinstatement during hits (i.e., encoding-retrieval similarity for correctly remembered same items) as compared to correct rejections (i.e., encoding-retrieval similarity between items during encoding and correctly identified novel items during retrieval). (B) Same analysis for items with high confidence responses.

No item-specific activity was observed in the hippocampus (hits versus correct rejections: all $p > 0.184$; high confidence hits versus high confidence correct rejections: all $p > 0.209$), further confirming our initial results (see figure below).

Hippocampus (n = 8)

Hits versus Correct Rejections

High Confidence Hits versus High Confidence Correct Rejections

Supplementary Figure S14: Comparison of reinstatement during hits as compared to correct rejections in the hippocampus. No significant differences were observed.

We report these new results in the novel Supplementary Figure S14.

We also compared reinstatement for hits versus correct rejections between congruent and incongruent spatial context conditions in the hippocampus and the LTC. For these analyses, we first subtracted hits and correct rejections for congruent and incongruent trials separately, and then statistically compared these contrast values. This was necessary because applying the surrogate-based cluster analysis to interactions (in this case: hits vs. correct rejection x congruent vs. incongruent) requires first computing contrasts in individual conditions, and then analyzing the interaction effect as the difference between the contrasts. In this interaction analysis (hits congruent minus correct rejections congruent versus hits incongruent minus correct rejections incongruent), we observed a significant cluster in a very similar time window as the cluster observed in the original contrast in the hippocampus ($p = 0.014$). This important novel result clearly supports our main conclusion that hippocampal reinstatement – now measured via the contrast of reinstatement during hits vs. correct rejections – is context-dependent. The result indeed supports some degree of item-specific reinstatement in the hippocampus, as suggested by the referee, but shows that this only occurs for congruent contexts.

Interaction analysis: hippocampus (n = 8)

Supplementary Figure S15: Interaction between hits, correct rejections and congruent – incongruent contexts in the hippocampus.

In the LTC, no significant cluster was observed in the interaction analysis (all $p > 0.206$). Again, this result underlines the difference in reinstatement between hippocampus (where reinstatement is context-dependent) and LTC (where it is not).

Interaction analysis: lateral temporal cortex (n = 11)

Supplementary Figure S16: Interaction between hits, correct rejections and congruent – incongruent contexts in the LTC.

We report these new results in the Supplementary Note 2 and Supplementary Figures S15 and S16.

The ERS in the HC appears quite early during retrieval (0 to 0.5s). This seems almost too early, and raises the concern of whether this is a visual response (e.g., the same visual response to the item and/or context upon seeing the item during encoding and retrieval). Again, this would be addressed if there were a proper comparison between correct and incorrect trials. Interestingly, in the HC the ERS does not distinguish same from different items or contexts, which may actually argue against a visual response.

Response: Thanks for highlighting this point. Indeed, in the previous version of the manuscript, we did not explicitly address the apparent short latencies in the hippocampal ERS contrast. In the revised manuscript, we have now added a discussion of this point. We agree that the results in the hippocampal "room" contrast lend further support to an interpretation of the observed ERS effects in terms of memory, according to which increases are dependent on the reinstatement of activity due to memory rather than being simply a consequence of the perceptual match between visual information at encoding and retrieval.

Notably, the time resolution of our ERS analyses was not optimized to assess the exact timing of the effects, but rather to maximize robustness and to be in line with previously published analyses (see for instance Staresina et al., *eLIFE* 2016; Zhang et al., *Nat Commun* 2018), i.e., feature vectors were built in windows of 500 milliseconds and sliding in steps of 100ms. We have added results of additional analyses at a 10-fold higher temporal resolution in order to assess the timing of ERS effects more accurately, both in the hippocampus and in the lateral temporal cortex (see the new Supplementary Note 4 and Supplementary Figure S17). This analysis indeed showed that hippocampal effects occurred only about 400ms after item onset, and LTC effects (in the behaviorally relevant cluster) after ~1600ms.

In the Supplementary Note 4, we now write:

"Hippocampal reinstatement of item-context associations occurred very early after the onset of item presentations during retrieval (0-500ms). This can be explained by several factors. First, in the experimental paradigm, participants have already entered a specific room in the virtual environment before any item is presented during retrieval – subjects have to reach the end of the rooms before being presented with items, which takes about 500 - 1000ms after entering the room (depending on navigation speed). Thus, contextual information is already available to them when retrieval starts. It is therefore likely that a contextual signal (that is present before item onset) modulates hippocampal ERS. Importantly, we would like to stress that this contextual signal alone cannot explain the effect of congruency on hippocampal ERS, as demonstrated in the contrast of same vs. different rooms (independent of items; see Figure 2C)."

Second, please note that we plot the results corresponding to a specific time window at the onset of individual time bins (see Methods). That is, activity marked at zero in the reinstatement maps includes signals ranging from 0 to 500 milliseconds, activity marked at 100ms includes activity between 100-600ms after onset etc. Given that the final calculation of ERS is applied to an entire 500ms time window, the apparent early onset may be due to increases in ERS occurring hundreds of milliseconds later.

In order to address the timing of ERS effects more specifically, we performed an additional analysis in which we increased the temporal resolution by a factor of 10, i.e., we constructed representational patterns in windows of 50 milliseconds, sliding with a 10ms increment. We specifically focused on temporal windows of 1.5 seconds duration centered around the effects that were observed previously (hippocampus: 1.5-3 seconds in encoding time, -0.5-1 seconds at retrieval; LTC: 1-2.5 seconds at encoding, 2-3.5 seconds at retrieval). These analyses revealed increases in hippocampal ERS after about 400ms ($p = 0.026$, Supplementary Figure S17, top). The onset of effects in the LTC, identified in the confidence contrast, were observed starting at ~1.5s after cue presentation and lasted until 1.9s, i.e., until the time before participants gave their responses ($p = 0.018$; Supplementary Figure S17, bottom)".

High Temporal Resolution Analysis

Hippocampus

Lateral Temporal Cortex

Supplementary Figure S17: Reinstatement effects at higher temporal resolution. Top row: hippocampal reinstatement of item-context associations. Bottom row: lateral temporal cortex reinstatement of item-specific information.

Nonetheless, an additional analysis could be performed in which the authors lock the data to the actual response. Presumably, any evidence of reinstatement should occur right before the subjects make their response, and would be more reassuring that this is related to the process of memory.

Response: While we agree in principle, our paradigm is less suited for detecting response-locked reinstatement than previous studies (e.g., see the paper by Yaffe et al., *PNAS* 2014). This is because (1) we did not ask participants to respond as fast as possible, and thus various factors may account for the variance of responses across trials – factors that are independent of the trial-specific timing of reinstatement; (2) additional variance of response times is added by the fact that the memory test in our experiment is integrated into a spatial navigation paradigm. These two factors may explain why we did not observe significant ERS effects in response-triggered analyses (see Figure below). Yaffe and colleagues (2014) also used an associative memory test, and the mean reaction time they observed is similar to ours (1831 vs. 1872ms for correct trials; 2736 vs. 2617ms for incorrect trials). However, the variability (STD) reported in their analyses is considerably smaller for both correct and incorrect trials (correct: 452 vs. 625ms; incorrect: 656 vs. 925ms), which may explain why this previous study observed significant condition effects of response-locked reinstatement which did not occur in our data (hippocampal congruent incongruent contrast: all $p > 0.433$; LTC item contrast: all $p > 0.373$; see Figure below). (Note that we converted variability from SEM as reported in their paper to STD in order to allow for a meaningful comparison with our results given the different numbers of subjects.)

These novel results are now presented in the Supplementary Note 9 and Supplementary Figure S18.

Hippocampus: response-locked analysis

Lateral Temporal Cortex: response-locked analysis

Supplementary Figure S18: Response-locked analyses. (A,B): Comparison of congruent and incongruent trials in the hippocampus (no significant cluster). (C,D): Comparison of same versus different items during encoding and retrieval (no significant cluster).

When looking at the interaction between HC ERS and LTC ERS, the authors perform a correlation between the two average values. But for this analysis, they examine the ERS in the LTC for same item, all contexts. If they are examining behaviorally relevant item specific reinstatement in the LTC, then they should correlate the difference in HC ERS between congruent and incongruent with the difference in LTC ERS between same and different items.

Response: In principle, we agree that it would be interesting to correlate the differences we observe in the different conditions detected with specific contrasts. However, since these contrasts are computed across trials of different conditions, the suggested correlation analysis would only have been possible *across participants*. This is arguably a less direct way of assessing relationships because inter-individual differences are driven by various factors including age, gender, cognitive status, etc.; and, in the case of patient groups: pathology. By contrast, we conducted correlation analyses *within subjects across trials*, excluding inter-individual variance.

This may not have been entirely clear in the manuscript, and we therefore described the analysis approach more explicitly (p. 50, l. 1081-1082). In addition, we are also including the following paragraph in Supplementary Note 7:

“Please note that these correlations were obtained across trials within individual participants. In principle, it would also have been interesting to correlate the magnitude of the respective ERS contrasts in HC and LTC. However, such a correlation of contrast values would only have been possible across participants, which is generally a less direct approach because inter-individual differences are driven by various different factors including age, gender, cognitive status etc.; and, in the case of patient groups: pathology. For these reasons we have not pursued that possibility.”

Indeed, we did not observe any inter-individual correlations between the magnitude of HC and LTC contrasts confirming our earlier choice (LTC same vs. different items, cluster ii: $\rho = -0.238$; $p = 0.582$; LTC high versus low confidence cluster: $\rho = 0$, $p = 1$; see figure below).

The analysis of phase synchrony is interesting, but seems quite complicated. They correlate levels of phase synchrony during different time windows with LTC ERS (which LTC ERS?). This is an indirect test. Why not directly examine phase synchrony between the two structures in the different conditions? Is there a difference between conditions, and if so, does then this difference correlate with the differences in ERS?

Response: We are sorry for the lack of clarity regarding the rationale and exact approach of this analysis. Indeed, we correlated phase synchrony during different time windows with LTC ERS in the behaviorally relevant cluster (identified in the confidence contrast). Again, this correlation analysis was conducted across individual trials, which in turn required us to obtain a single-trial measure of phase synchrony. We thus analyzed phase synchrony across time (within the individual trials).

This is now described in greater detail on p. 12, l. 283-290 of the manuscript (please see also the description in the Methods section on p. 51, l.1122-1126).

In addition, we now also conducted a more conventional phase synchrony (PLV) analysis, i.e., we analyzed PLV across trials and compared the magnitude of PLV between the different conditions. Note that PLV is a measure that is sensitive to the number of trials and we therefore matched trial numbers across conditions by randomly removing trials from the condition with more trials to match the condition with fewer trials for each subject independently. PLV did not differ between congruent and incongruent or between high confidence versus low confidence trials (all clusters $p > 0.154$).

Hippocampal - Lateral Temporal Cortex PLV analysis

Hippocampal - Lateral Temporal Cortex PLV analysis

In addition, why does the absence of a correlation between synchrony and HC ERS mean that there is no recurrent influence from LTC to HC? This does not address the direction of influence, and the timing alone of the respective ERS would suggest that this should not occur.

Response: We agree with the reviewer that this statement was misleading. Indeed, our finding that ERS effects in HC were substantially earlier than in the LTC by itself suggests that the direction of influence is from the hippocampus to the LTC. In the revised version of the manuscript, we have removed the sentence “suggesting that HC coordinates the LTC response with no recurrent influence from LTC upon HC dynamics”.

Methodologically, all of the analyses are based on examining the time series data from one electrode contact each in the HC and LTC in each patient. The ERS is calculated based on this time series. But how is this one electrode chosen in each patient? Surely some patients have more than one electrode placed in the hippocampus, for instance, which means they have more than one set of two 'deepest' contacts to choose from. Same with the LTC. What do the data look like for other electrode choices?

Response: We apologize that the electrode selection criteria were not sufficiently clear in the original manuscript. Indeed, we selected only one electrode contact in the hippocampus and one in the LTC of each patient, in line with previous studies (Staresina et al., *eLife* 2016; "temporal RSA" approach in Zhang et al., *Curr Biol* 2015). Our patients never had more than one electrode targeting the anterior hippocampus. From this electrode, we chose the most distal hippocampal contact and the contact that was located most centrally in gray matter in the lateral temporal cortex. Across all our patients, we used a contact from the same electrode as the hippocampal contact in order to minimize variance in spatial location of the LTC contact. This is now described on p. 42, l. 885-913 (Methods section). We decided to select these two areas because of the well-documented relevance of the anterior hippocampus for relational memory (e.g., Eichenbaum, *Annu Rev Neurosci* 2007) and more specifically, based on the finding of increased reinstatement for source as compared to item memory in the anterior hippocampus (Staresina et al., *eLife* 2016). The LTC has been previously linked to recognition memory and has been investigated in similar iEEG setups (Jang et al., *Curr Biol* 2017). However, no previous study directly compared reinstatement in hippocampus and neocortex.

As the Reviewer points out, from this one electrode sometimes more than one contact reached the hippocampus or was located in LTC grey matter (in the anterior and posterior regions of the HC, for instance, or on two sides of the same gyrus in the LTC). After co-registering pre- and post-electrode placement using MR images and CT whole-brain volumes, we could confirm a total of 15 contacts located in the anterior hippocampus and 10 contacts in the posterior hippocampus across the group (excluding patients with hippocampal epilepsy).

In order to check for the robustness of our results, we performed an additional analysis in which we included all available electrodes in each specific ROI (please see response to next comment below).

Was there any consideration given the performing the analyses here using multiple contacts in the HC and the LTC, constructing ERS based on the time-space patterns of activity across multiple contacts?

Response: Following previous research (Staresina et al., *eLife* 2016; Zhang et al., *Curr Biol* 2015), we initially did not consider constructing representational patterns of more than one electrode. However, following the suggestion of the reviewer as well as comments by reviewer 2, we conducted several additional analyses:

- We calculated RSA at one posterior hippocampal contact in each patient.
- We calculated hippocampal RSA including activity from all available hippocampal contacts in each patient (total: n=25 contacts). The MNI locations of included electrodes in this analysis are presented in Table 1.
- We built LTC RSA patterns from the activity of all contacts located in grey matter in the lateral temporal lobe in each patient (total: n=117 contacts; see also Table 2 for details).

These complementary analyses show that hippocampal reinstatement is mostly driven by its anterior part. Indeed, no clusters survived multiple comparisons correction in the posterior hippocampal analysis or in the analysis including all hippocampal contacts (all $p > 0.180$; see panels A and D in the Supplementary Figure below). However, when focusing on the previously identified temporal hippocampal cluster, we found a significant difference between congruent and incongruent trials in the multielectrode analysis ($t(7) = 2.397$; $p = 0.048$; see Supplementary Figure S4 below, panel E). While this difference was not observed in the posterior hippocampus, ERS was significantly higher than zero also in that region for congruent but not for incongruent trials (congruent: $t(5) = 2.798$; $p = 0.038$; incongruent: $t(5) = 0.397$; $p = 0.707$; see Supplementary Figure S4 below, panel B).

Posterior Hippocampus (n = 6)

Combined anterior and posterior hippocampus (n = 8)

Supplementary Figure S4: ERS in the posterior hippocampus (top) and in combined anterior and posterior hippocampus (multi-electrode analysis; bottom). (A) Lack of congruency effect in posterior hippocampus (B) Analysis of congruency effects in previously identified cluster from the anterior hippocampus showing significant encoding-retrieval similarity for congruent but not for incongruent items. (C) Lack of item-specific reinstatement in the posterior hippocampus. (D) Lack of

congruency effect in the multi-electrode analysis. (E) Analysis of congruency effects in previously identified cluster from the anterior hippocampus, now including concatenated activity from all available hippocampal contacts. Plot shows significant differences in encoding-retrieval similarity for congruent as compared to incongruent items. (F) Lack of item-specific reinstatement in the multielectrode analysis.

Results of the multi-electrode analysis in the LTC revealed no item or context-specific reinstatement (all $p > 0.109$), suggesting a localized representation of items in more anterior regions of the temporal lobe where the electrodes of the main analysis are located. In the confidence contrast, we observed a significant increase of ERS for high as compared to low confidence trials starting 2 seconds after cue onset in encoding and 3 seconds in retrieval time ($p = 0.035$).

Lateral Temporal Cortex global reinstatement analysis (n = 11)

A

B

Supplementary Figure S5: Reinstatement of item-specific information based on multi-electrode patterns in lateral temporal cortex. (A) All available electrodes in the LTC of each patient are plotted in common MNI space. Cyan circles indicate locations of electrodes that were chosen in each subject for the main (single-electrode) LTC analysis. (B) Results from the multi-electrode (global) ERS analysis: significant reinstatement for high versus low confidence trials; lack of significant congruency or item-specific effects.

Reviewer #2 (Remarks to the Author):

The authors propose to investigate the theory of human episodic memory that proposes that detailed sensory representations of individual items rely on the neocortex whereas the hippocampus realizes an index to these cortical representations through item-context associations. This theory predicts information flow from neocortex to hippocampus during memory formation and the reverse direction during memory retrieval.

The authors study patients with intractable epilepsy implanted with intracranial EEG in the hippocampus and lateral temporal cortex for clinical purposes in 7 patients. They used one hippocampal and one lateral temporal cortex electrode from the same hemisphere in each patient. They excluded 2 patients because the seizure onset zone was in the unilaterally implanted hippocampus. However, some patients had electrodes in the left and others in the right hemisphere. Subjects performed a Virtual Reality active navigation task involving a recognition memory test (congruent and incongruent conditions).

They calculated ERS (Encoding-Retrieval Similarity) between feature vectors and found significantly higher ERS in congruent versus incongruent trials. They did not find evidence for reinstatement of item-specific information in the hippocampus. In addition, they found that ERS values were significantly higher than zero in congruent trials but not incongruent trials.

In the lateral temporal cortex they found item specific ERS values with 2 time windows during encoding that are reinstated during a retrieval time window between 1.5-2.5 seconds. In contrast to findings in the hippocampus they did not observe reinstatement of item-context associations.

They also investigated the behavioral relevance of their findings and did not detect a difference in task performance although the participants recalled most items. They investigated high confidence versus low confidence. Higher confidence trials were associated with higher levels of reinstatement in the left temporal cortex and not the hippocampus. Based upon these findings the authors conclude that item representations in lateral temporal cortex and item-context associations in the hippocampus and also temporally with item-context (hippocampus) occurring earlier during retrieval than item reinstatement (lateral temporal cortex).

In the hippocampus the delta/theta bands were found to be significant in reinstatement of item-context associations. There was not item-specific frequency band in the left temporal cortex. As the authors point out, these findings are in contrast to prior ECoG studies in the hippocampus.

The authors also investigated a possible mechanism underlying coordination of reinstatement – phase synchronization. They observed a correlation in lateral temporal cortex ERS and Hippocampus-lateral temporal cortex phase synchronization from 70-82 Hz (gamma) but no correlation in the opposite direction (from lateral temporal cortex to hippocampus) suggesting the hippocampus coordinated lateral temporal cortex response.

Patients: all had unilateral implantations of stereotactic EEG with 7-10 electrodes with 5-15 contacts. In figure 3 the lateral temporal electrodes appear to be fairly closely localized in an axial plan but more detail is needed. Figure 1: From figure one it appears that there were 3 right sided and 4 left sided electrode contacts included.

The study was performed rigorously and methods are well described and logical. The findings are novel.

Response: We thank the referee for the overall positive comments on our manuscript. Following up on the reviewer's request, we now present more information related to electrode locations,

including coordinates in MNI space for each contact in Supplementary Table 2. In addition, we also present figures with lateral convexity views in specific analyses, where electrodes are plotted superimposed on a structural MRI (e.g., see Fig. 1 and Supplementary Figures S5 and S6 for the multi-electrode analyses). Indeed, there were 3 patients with right sided and 4 patients with left sided electrodes in the original group, and 4 and 7 respectively in the n=11 group. This is now also clearly stated in the manuscript (p. 42, l. 888).

However, the number of subjects included in the study are limited and the authors only present data from the electrodes in the lateral temporal cortex and anterior hippocampus.

Response: We agree that the number of subjects included in our study appears relatively low, however it is similar to the number of patients in previous intracranial EEG studies (see for instance the studies by Voytek et al., *Nat Neurosci* 2017 (4 patients); Vass et al., *Neuron* 2016 (3 subjects); or Ekstrom et al., *Hippocampus* 2005 (6 patients)). In general, the relatively low number of subjects is compensated by the superior signal to noise ratio of iEEG as compared to fMRI or scalp EEG data.

Nevertheless, in order to avoid possible concerns related to the number of subjects, we performed additional actions:

- 1) We included additional data from 2 patients who were originally discarded because they were diagnosed with hippocampal epilepsy on the side of electrode implantation. In these cases, we only focused analyses on the LTC electrodes.
- 2) We collected data from 2 further patients. One of them was eventually diagnosed with anterior hippocampal epilepsy and only had unilateral hippocampal implantations, whereas the other patient had a seizure onset zone outside of the hippocampus.

In total, we could thus include data from 11 subjects. Some of them had to be excluded from specific analyses. For instance, all subjects with a hippocampal seizure onset zone were excluded from analyses involving the hippocampus. On the other hand, all 11 subjects were included in the LTC contrasts, but as an additional precaution, we verified the results also in the group of n=8 subjects without hippocampal epilepsy. We now specify in the Methods section of the updated manuscript on p. 39, l. 812-821 the number of subjects that were included. For their demographic and clinical data, please see also Supplementary Table 1.

The inclusion of these additional patients corroborated our main findings. With regard to the hippocampus, across our new group of 8 subjects, the main results observed in the HC contrasts remain present: We still observed a significant increase for reinstatement of congruent as compared to incongruent stimuli in the same time window and no difference for ERS of same vs. different items or same rooms vs. different rooms. We updated all respective figures and statistical values in the manuscript.

With regard to the LTC, we confirmed our original finding of item-specific reinstatement (in the absence of congruency-dependent reinstatement) in the larger sample of n=11 subjects as well as in a sample of n=8 subjects (excluding those with ipsilateral hippocampal seizure onset). In both groups, we found 2 reinstatement clusters in the same time periods as in our original analysis (bottom line of the figure below).

In the high versus low confidence contrast, we only included subjects who responded with low confidence (n=8 participants in our extended group and n=7 when excluding patients with ipsilateral hippocampal seizure onset). In these larger groups, we observed a significant cluster of

increased ERS for high as compared to low confidence trials in a very similar time period as in the initial analysis.

Hippocampus (N = 8)

Lateral Temporal Cortex (N = 8)

Lateral Temporal Cortex (N = 11)

Novel main figures: Confirmation of main findings in extended groups of patients.

In our frequency-specific analysis, we further confirmed the contribution of low-frequency oscillations to hippocampal, but not LTC, reinstatement: When focusing on the relevant temporal regions of interest of each region, we observed that mean ERS in the hippocampal congruency cluster was significantly reduced, while ERS in the LTC clusters from the item or confidence contrasts did not change. Note that clusters in this analysis were identified based on our new group of n=8 subjects (i.e., those without hippocampal epilepsy).

Figure 4: Frequency-specific reinstatement in hippocampus and lateral temporal cortex

(A) Results from jackknife procedure showing significant reduction of hippocampal context reinstatement after excluding 1-8Hz activity, while item-specific reinstatement and confidence effects in lateral temporal cortex were not affected. Bar plots indicate changes in encoding-retrieval similarity in the main contrasts and clusters after excluding 1-8Hz activity. * in panel A indicates reductions in ERS values that are significantly different from 0 across the group at $p < .05$. **(B,C)** Reduction of encoding-retrieval similarity in relevant temporal regions of interest after the removal of delta (1-3Hz; panel A) and theta (4-8Hz; panel B) activity. While reductions were numerically higher in the hippocampal congruent-incongruent cluster, no significant differences were observed.

In the coordinated reinstatement analysis, we only included patients without hippocampal epilepsy who showed behavioral responses with both high and low confidence ($n=7$). This is because the temporal regions of interest in the LTC tROI analysis (Figure 5A) and in the sliding time windows approach (Figure 5B and 5C) were defined based on the cluster observed in the confidence contrast. Our results remained quantitatively the same in this larger group: We still observed significant coordinated reinstatement across our group of participants (Figure 5A), which was specific to the tROIs showing content-specific and functionally relevant reinstatement. We have updated all figures and statistical values in the manuscript to reflect these updated results.

Figure 5: Coordination of representational reinstatement between hippocampus and lateral temporal cortex.

(A) Encoding time-matched temporal regions of interest (tROIs) were defined based on the significant clusters supporting reinstatement of item-context associations in the hippocampus and confidence effects in the lateral temporal cortex. Reinstatement values were correlated across trials in each patient. **(B)** Correlation map between the hippocampal tROI cluster and all encoding-retrieval time pairs (0-3s) in the lateral temporal cortex (cluster equally sized as in panel A, top right). **(C)** Correlation map between the LTC cluster and all encoding-retrieval time pairs in the HC. Bottom parts of panels B and C show T-maps in which correlations against zero at the group level were tested for the correspondent analysis. Areas in which values reached significance at $p < .05$ (uncorrected) are outlined.

The results in the phase synchrony analysis in our extended group again revealed a significant correlation between single trial levels of LTC confidence reinstatement and Phase Locking Value (PLV) between the hippocampus and the LTC in the 77-83Hz frequency range during the period of hippocampal reinstatement (0-500ms after cue onset, $p = 0.023$, see panel A in the figure below). This effect was not observed when we correlated early phase synchronization with mean hippocampal ERS in the congruency cluster (all $p > 0.131$, panel B in the figure below).

Figure 6: Gamma phase synchronization between hippocampus and lateral temporal cortex correlates with lateral temporal cortex reinstatement.

(A) Frequency-specific (77-83Hz) hippocampal-lateral temporal cortex phase synchronization significantly correlates with behaviorally relevant reinstatement in lateral temporal cortex (blue transparent box). (B) Hippocampal-lateral temporal cortex phase synchronization does not correlate with hippocampal reinstatement during the time period of increased hippocampal ERS. Curves and shaded areas in A and B show mean + SEM of correlation of phase synchronization and LTC reinstatement (red) and mean + SEM of baseline-corrected hippocampal-LTC early phase synchronization (0-500ms at retrieval, black).

In addition, roughly half of the subjects have left implantations and half have right implantations. There is no discussion regarding the limitations of having a mix of right and left sampling and the authors do not present results of each hemisphere individually.

Response: We agree that lateralization is a potentially relevant issue. However, it is a common practice in intracranial EEG studies to aggregate data collected from different hemispheres to increase statistical power, both within and across participants; e.g., see Méndez-Bértolo et al., *Nat Neurosci*, 2016; Yaffe et al., *PNAS*, 2014; Staresina et al., *eLife*, 2016; Fell et al., *Nat Neurosci*, 2001. Since previous studies showed evidence for a bilateral contribution of medial temporal lobe structures to spatial memory (Glikmann-Johnston et al., *Brain* 2008; Shinohara et al., *Hippocampus* 2012; Burgess et al., *Neuron* 2002), we did not expect pronounced lateralization effects. This is indeed what our results reveal. In the hippocampus, we did not find context reinstatement in either the left (n=5) or the right (n=3) hemisphere groups. In the LTC, while we observed item-specific effects in the left hemisphere group only (n= 7), a direct comparison of contrast values in the previously identified clusters revealed no statistical differences between the two groups ($t(9) =$

0.044, $p = 0.965$, unpaired t-test). In the revised version of the manuscript, we now report the results of the lateralization analyses in the Supplementary Note 6 and the Supplementary Figures S7-S8.

In Supplementary Note 6, we state: “In order to specifically test possible influences of lateralization, we assessed reinstatement in our main contrasts by only including patients with left or right implantations (results were Bonferroni corrected for the 2 comparisons in the separate hemispheres). In the hippocampus ($n=5$ left, $n=3$ right), we did not find significant differences between congruent and incongruent items in either the left hemisphere (all $p > 0.258$) or the right hemisphere group ($p = 0.07$). In the LTC ($n=7$ left, $n=4$ right), item-specific effects were observed in the left hemisphere around the same time of cluster ii in the main analysis ($p = 0.029$). In the right hemisphere group, no differences were observed between conditions (all $p > 0.095$).

In addition, we directly compared reinstatement patterns between patients with electrodes in the left and right hemisphere. For this analysis, we focused on contrast values in the previously identified clusters (see Methods). Mean ERS values in the hippocampal cluster were not statistically different between hemispheres ($t(7) = -0.31$, $p = 0.761$, unpaired t-test). In the LTC, we compared mean ERS values between left and right hemisphere groups in cluster ii of the main analysis (i.e., where the item and behaviorally relevant effects were found). Again, no statistical difference was observed ($t(9) = 0.044$, $p = 0.965$, unpaired t-test).”

In the Discussion (p. 18, l.420-431), we write:

“Even though it is a common practice to aggregate data collected from different hemispheres to increase statistical power in intracranial EEG studies, we compared reinstatement patterns in patients with electrodes in the left and right hemisphere. No difference was observed in the hippocampus; in LTC, the left but not the right hemispheric group showed an effect. A direct comparison did not reveal any significant hemispheric differences. The lack of pronounced lateralization effects may be explained by the contribution of medial temporal lobe structures of both hemispheres to memory (Glikmann-Johnston et al., 2008; Shinohara et al., *Hippocampus* 2012; Burgess et al., *Neuron* 2002). In this context, it should be noted that the prevalence of atypical language lateralization is higher in epilepsy patients than in the general population (Helmstaedter et al., 1997), which may obscure possible lateralization effects if language lateralization is not explicitly assessed. This is because in patients with atypical language lateralization, material-dependent memory functions often shift their hemispheric distribution as well (Helmstaedter et al., 2006).”

Lateralization: Hippocampus

Left hemisphere (n = 5)

Right hemisphere (n = 3)

Left versus Right hemisphere (n = 8)

Supplementary Figure S7: Analysis of hippocampal reinstatement in left and right hemispheres. No significant reinstatement was observed when analyzing the two patient groups separately. A direct comparison of mean ERS values within the hippocampal congruent – incongruent cluster revealed no statistical differences between the two groups.

Lateralization: Lateral Temporal Cortex

Left hemisphere (n = 7)

Right hemisphere (n = 4)

Left versus Right hemisphere (n = 11)

Supplementary Figure S8: Analysis of lateral temporal cortex reinstatement in left and right hemispheres. Significant item-specific reinstatement was observed in the left hemisphere group, while this effect was not present in the right hemisphere group. A direct comparison of mean ERS values within the item-specific LTC cluster revealed no statistical differences between the two groups.

Another major criticism related to the lack of control electrodes analyzed. Each subject had 5-15 electrode contacts implanted. I suggest that the authors do additional analysis in a non-lateral temporal lobe cortical region to ascertain whether the findings are specific to the lateral temporal lobe relationship with the anterior hippocampus. For instance, use a parietal cortical contact pair as a control.

Response: We agree that this is a relevant control analysis. In addition to the anterior HC and LTC, we now report data from posterior hippocampal contacts (n = 10); across all anterior and posterior hippocampal contacts (n = 25 contacts); across multiple contacts in the lateral temporal lobe (n =

117); and from the parietal lobe ($n = 19$ contacts; see also Table 2). In all of these novel analyses, except for the posterior hippocampus analysis, we built representational patterns based on concatenated activity from more than one contact, i.e., from all electrodes in a given region of interest available for each subject. We describe in the Methods section (l. 961-971, p. 45) how the representational patterns were constructed by concatenating time-frequency profiles of different electrodes before conducting the similarity comparisons. We also refer to the methods we used to identify electrode locations in the updated manuscript (l. 885-913).

These complementary analyses show that hippocampal reinstatement is mostly driven by its anterior part. Indeed, no clusters survived multiple comparisons correction in the posterior hippocampal analysis or in the analysis including all hippocampal contacts (all $p > 0.180$; see panels A and D in the Supplementary Figure below). However, when focusing on the previously identified temporal hippocampal cluster, we found a significant difference between congruent and incongruent trials in the multielectrode analysis ($t(7) = 2.397$; $p = 0.048$; see Supplementary Figure S4 below, panel E). While this difference was not observed in the posterior hippocampus, ERS was significantly higher than zero also in that region for congruent but not for incongruent trials (congruent: $t(5) = 2.798$; $p = 0.038$; incongruent: $t(5) = 0.397$; $p = 0.707$; see Supplementary Figure S4 below, panel B).

Posterior Hippocampus (n = 6)

Combined anterior and posterior hippocampus (n = 8)

Supplementary Figure S4: ERS in the posterior hippocampus (top) and in combined anterior and posterior hippocampus (multi-electrode analysis; bottom). (A) Lack of congruency effect in posterior hippocampus (B) Analysis of congruency effects in previously identified cluster from the anterior hippocampus showing significant encoding-retrieval similarity for congruent but not for incongruent items. (C) Lack of item-specific reinstatement in the posterior hippocampus. (D) Lack of congruency effect in the multi-electrode analysis. (E) Analysis of congruency effects in previously identified cluster from the anterior hippocampus, now including concatenated activity from all available hippocampal contacts. Plot shows significant difference in encoding-retrieval similarity for congruent as compared to incongruent items. (F) Lack of item-specific reinstatement in the multielectrode analysis.

Results of the multi-electrode analysis in the LTC revealed no item or context-specific reinstatement (all $p > 0.109$), suggesting a localized representation of items in more anterior regions of the temporal lobe where the electrodes of the main analysis are located. In the confidence contrast, we observed a significant increase of ERS for high as compared to low confidence trials starting 2 seconds after cue onset in encoding and 3 seconds in retrieval time ($p = 0.035$).

Lateral Temporal Cortex global reinstatement analysis (n = 11)

A

B

Supplementary Figure S5: Reinstatement of item-specific information based on multi-electrode patterns in lateral temporal cortex. (A) All available electrodes in the LTC of each patient are plotted in common MNI space. Cyan circles indicate locations of electrodes that were chosen in each subject for the main (single-electrode) LTC analysis. (B) Results from the multi-electrode (global) ERS analysis: significant reinstatement for high versus low confidence trials; lack of significant congruency or item-specific effects.

In the parietal cortex, we did not find any difference between conditions in the context (all $p > 0.110$) or item comparisons (all $p > 0.129$). Interestingly, however, we observed increased reinstatement for high versus low confidence trials around 2 seconds after cue presentation in encoding and retrieval time ($p = 0.011$; see below), suggesting an involvement of the parietal cortex in the encoding of subjective confidence. These findings are consistent with the well-known role of the posterior parietal cortex in the processing of subjective memory confidence (Simons et al., *Cereb Cortex* 2010; Rutishauser et al., *Neuron* 2018; Wynn et al., *Learn Mem* 2018). Note that these analyses were performed on a group of $n=5$ subjects (including all subjects with parietal electrodes).

Parietal Cortex global reinstatement analysis (n = 5)

A

B

Supplementary Figure S6: Reinstatement effects in the parietal lobe. (A) Global RSA analysis was performed with concatenated activity from all available parietal electrode contacts in each patient, here shown in common MNI space. Each color represents a different subject. (B) Lack of reinstatement of item-context associations (Left), or item-specific information (Middle). Right: more pronounced encoding-retrieval similarity for items rated with high as compared to low confidence.

These additional results have been included to our manuscript in the novel Supplementary Note 5 and the novel Supplementary Figure S6.

In order to investigate whether our results in the coordinated reinstatement analysis (Figure 5) are specific to the relationship of the anterior hippocampus with the LTC, we calculated “coordinated

reinstatement” between the hippocampus and the posterior parietal cortex (PPC) as a control. As in the HC-LTC interaction analysis, we correlated single trials levels of hippocampal congruent-incongruent reinstatement and PPC confidence reinstatement in their respective temporal regions of interest (hippocampus: congruent incongruent cluster, PPC: confidence cluster, see above). In the ERS same item, all contexts condition, we found a non-significant relationship ($t(4) = 0.135$; $p = 0.89$), suggesting that the hippocampus specifically coordinates with the LTC during the retrieval of episodic memories. However, we note that the relatively low number of subjects with parietal electrodes and the higher variance in the spatial position of parietal contacts as compared to the LTC (see Supplementary Figure S6) might have affected these results. In addition, we would like to stress the need of further exploring the coordination of different forms of reinstatement across the brain (e.g., between the hippocampus and the prefrontal cortex, given the well-known involvement of the latter in memory retrieval during systems consolidation; see Eichenbaum, *Annu Rev Neurosci* 2017).

We have included the results of the interaction analysis in the novel Supplementary Note 10 and the novel Supplementary Figure S20

Finally, there is very little detail regarding the patient characteristics (e.g. age, seizures onset region on ECoG, gender, etc.) This should be included.

Response: We are now providing a summary table (see Supplementary Table 1) which includes all the necessary information about the patients.

Minor point:

Line 171: lower- confidence should be low confidence

Response: This has been changed.

Reviewer #3 (Remarks to the Author):

In this paper, Pacheco and colleagues report an intracranial EEG experiment investigating memory reinstatement in hippocampus and lateral temporal cortex (LTC). They recorded simultaneously from both regions, allowing them to relate hippocampal reinstatement to LTC reinstatement effects, which were temporally related and correlated across trials. Interestingly, hippocampal effects were modulated by the match of both item and context information at encoding and retrieval, whereas LTC effects were modulated only by item match, consistent with models of hippocampal function that emphasize contextual binding processes.

In general, I thought the paper was timely, theoretically interesting, and important in its advances beyond the existing literature. The use of intracranial EEG allowed for more sophisticated evaluation of the temporal relationship between different aspects of reinstatement than has been previously documented (e.g., with fMRI or with EEG studies recording from a single region). Such hippocampal-neocortical interactions factor heavily in neurobiological models of memory. The paper was also well-written. I have relatively few comments.

Response: We thank the reviewer for the positive evaluation of our work and suggestions for improvement. Please see below for our responses.

The authors should elaborate further on their decision to group the delta and theta bands. It would seem warranted to report the results for delta and theta separately (in addition to together, which does seem motivated by the literature).

Response: We agree with the reviewer that the justification we provided in the initial version of the manuscript (arguing that there is an “ongoing discussion about the human correlate of rodent theta oscillations, in particular during spatial navigation^{51,73,74”}) should be elaborated.

We now provide a more detailed explanation of why we combined theta and delta oscillations in the Methods (p. 48, l. 1034-1053):

“Theta oscillations (about 4-8Hz) comprise a functionally relevant frequency band which has been linked in the animal literature to important cognitive processes such as active learning, memory encoding, and spatial navigation (e.g., Buzsaki, *Neuroscience* 1989; *Neuron* 2002; Colgin, *Nat Neurosci* 2016). A similar function of this frequency band has been hypothesized to exist in humans, albeit at slightly slower frequencies (2-3Hz, or delta; Watrous et al., *J Neurophysiol* 2011; Lega et al., *Hippocampus* 2012; Ekstrom, *Hippocampus* 2005). Based on these results, it has been suggested that the frequency of human theta oscillations is lower than in rodents (Jacobs, *Philos Trans R Soc Lond B Biol Sci* 2014), possibly due to the larger anatomical extent of neural assemblies in humans as compared to rodents (Buzsaki and Draguhn, *Science* 2004). On the other hand, a recent study compared virtual and actual physical navigation and described theta oscillations at a higher frequency during real world as compared to virtual navigation, even though theta oscillations at a lower frequency occurred as well (Bohbot et al., *Nat Commun* 2017). Thus, similar to rodents (e.g., Bland, *Prog Neurobiol* 1986), there may be multiple theta generators in the human hippocampal formation that have different frequency profiles and distinct – sometimes even opposing (e.g., Oehrns et al., *Curr Biol* 2018) – functional roles. In the current study, we used a parsimonious approach and analyzed the contribution of activity across an extended frequency range including both conventional delta and theta oscillations (combined band: 1-8Hz).”

In addition, we now also separately assessed the contribution of 1-3Hz vs. 4-8Hz oscillations. We did not find a specific contribution of either the delta or the theta band to reinstatement in the hippocampus or the LTC. Although we still observed ERS increases around the same temporal

region where we found an effect in the original hippocampal 1-8Hz band analysis, this cluster did not survive correction for multiple comparisons ($p = 0.138$) in our extended group of $n=8$ subjects (please note that we increased our sample size following comments by reviewer 2). Therefore, we removed the analysis of band-limited reinstatement from the manuscript. Nevertheless, we confirmed the relevance of low-frequency oscillations to hippocampal reinstatement in our jackknife analysis, where we found a significant reduction of mean ERS in the congruent-incongruent hippocampal cluster after the removal of the 1-8Hz band. No such effect was observed in the LTC. We have updated Figure 4 of the manuscript with these novel results.

Figure 4: Frequency-specific reinstatement in hippocampus and lateral temporal cortex

(A) Results from jackknife procedure showing significant reduction of hippocampal context reinstatement after excluding 1-8Hz activity, while item-specific reinstatement and confidence effects in lateral temporal cortex were not affected. Bar plots indicate changes in encoding-retrieval similarity in the main contrasts and clusters after excluding 1-8Hz activity. * in panel A indicates reductions in ERS values that are significantly different from 0 across the group at $p < .05$. **(B,C)** Reduction of encoding-retrieval similarity in relevant temporal regions of interest after the removal of delta (1-3Hz; panel A) and theta (4-8Hz; panel B) activity. While reductions were numerically higher in the hippocampal congruent-incongruent cluster, no significant differences were observed.

It is unclear what is meant by “In order to exclude that effects were driven by global and content-unspecific event-related effects during encoding...” (lines 235-238), or how the analysis controlled for that possibility.

Response: We apologize for the lack of clarity. In the revised manuscript, we elaborate more on the rationale behind our analysis approach when assessing coordinated reinstatement in the novel Supplementary Note 7:

“Since our aim in the coordinated reinstatement analysis was to investigate the interaction (or “coordination”) of the different representational formats of reinstatement in the hippocampus and the LTC, we defined temporal regions of interest in each area that were aligned in time during encoding. Specifically, we only included in our analysis those encoding/retrieval time pairs that satisfied two conditions: First, significance in each region’s contrast; and second, identical encoding time. The rationale behind this approach was that coordinated reinstatement reflects activity from the same encoding period. Thus, we specifically investigated coordinated reinstatement during the time window in which we had observed content-specific reinstatement in the two individual regions of interest. In order to further validate the specificity in time of this interaction, we also analyzed coordinated reinstatement across all possible encoding/retrieval time windows. This analysis showed that coordinated reinstatement across regions is indeed specific to those time windows where reinstatement in the two individual regions occurs (Figure 5).”

A critical test of the validity of our decision on the temporal regions of interest (tROIs) in the coordinated reinstatement analysis is reported in the “coordinated reinstatement analysis” section. On p. 11, l. 250-256, we describe that we obtained the same pattern of results in these tROIs as in the original clusters (i.e., higher reinstatement for congruent vs. incongruent trials in the hippocampus and for same items vs. different items in the LTC).

In order to increase clarity, we rephrased this sentence in the manuscript (p.11, l. 250-251). We now write: “Because coordinated reinstatement is more likely to occur when it reflects activity from the same encoding time...”

The correlation of hippocampal and LTC reinstatement was conducted over trials for ERS_same-item, all-contexts. But the hippocampal effect was specific to the same-item, same-context comparison (vs same-item, different-context). Shouldn’t the correlation be limited to these trials? How many correlations were computed, and were they corrected for multiple comparisons? This issue is mentioned in the Discussion (lines 339-340) but it should be unpacked further.

Response: We agree that this is a relevant point. In the coordinated reinstatement analysis, we started with the most global analysis – i.e., including all trials that matched in terms of item content, regardless of context. Even though one may expect coordinated reinstatement specifically for the subset of trials in which also the contexts match, this was not the case, possibly due to reduced statistical power. This is now reported in greater detail in the Results section (l. 257-265).

With respect to the number of correlations, we first identified the relevant tROIs in each region and then performed a correlation of averaged ERS values from these regions. Since this analysis is based on a single test (presented in Figure 5A), we did not correct for multiple comparisons. For the sliding time window analyses, we run a total of 33*29 Spearman’s correlations tests in the hippocampal “seed” analysis (Figure 5B) and 34*35 in the analysis based on the LTC confidence cluster (Figure 5C). These results were corrected for multiple comparisons using a cluster-based

permutation statistics (see Methods, I. 1146-1154, p.52-53). This correction resulted in a corrected p-value of $p=0.087$, which we did not consider significant (although one may argue that a one-tailed test is justified when comparing empirical results to surrogates). Nevertheless, this result shows that coordinated reinstatement is not a general phenomenon that occurs unspecifically across all encoding/retrieval time windows. Instead, it is specific for the temporal regions of interest in which reinstatement effects were found in HC and LTC.

Figure 6 - I found it confusing that the black and red lines were displayed on the same plot, since they seem to refer to different types of measures.

Response: We have changed this and now show the results in two different plots in Figure 6.

REVIEWERS' COMMENTS:

Reviewer #1 (Remarks to the Author):

The authors have provided a significant number of analyses to address the concerns that have been raised by the reviews. I think providing these analyses has substantially improved the manuscript, and will make any interpretations of the results more clear for the reader.

Reviewer #2 (Remarks to the Author):

The authors have thoroughly addressed all of the points that I made. The additional analysis that they have completed add significantly to the paper and importance of this work.

One minor alteration:

In the supplemental table providing patient information they should have the header "MRI" or "Brain MRI" instead of RM. I believe this is a typo.

Reviewer #3 (Remarks to the Author):

The authors have completed a thorough revision, and they have sufficiently addressed all of my previous comments.